# Tonic interferon restricts pathogenic IL-17-driven inflammatory disease via balancing the microbiome

Isabelle J Marié[1]*, Lara Brambilla[1], Doua Azzouz[1], Ze Chen[1], Gisele V Baracho[2], Azlann Arnett[3], Haiyan S Li[4], Weiguo Liu[1], Luisa Cimmino[5], Pratip Chattopadhyay[1], Gregg Silverman[1], Stephanie S Watowich[4], Bernard Khor[3], David E Levy[1]*

[1]NYU School of Medicine, New York, United States; [2]BD Life Sciences, La Jolla, United States; [3]Translational Immunology, Benaroya Research Institute at Virginia Mason, Seattle, United States; [4]University of Texas MD Anderson Cancer Center, Houston, United States; [5]Sylvester Comprehensive Cancer Center, University of Miami Miller School of Medicine, Miami, United States

**Abstract** Maintenance of immune homeostasis involves a synergistic relationship between the host and the microbiome. Canonical interferon (IFN) signaling controls responses to acute microbial infection, through engagement of the STAT1 transcription factor. However, the contribution of tonic levels of IFN to immune homeostasis in the absence of acute infection remains largely unexplored. We report that STAT1 KO mice spontaneously developed an inflammatory disease marked by myeloid hyperplasia and splenic accumulation of hematopoietic stem cells. Moreover, these animals developed inflammatory bowel disease. Profiling gut bacteria revealed a profound dysbiosis in the absence of tonic IFN signaling, which triggered expansion of $T_H17$ cells and loss of splenic $T_{reg}$ cells. Reduction of bacterial load by antibiotic treatment averted the $T_H17$ bias and blocking IL17 signaling prevented myeloid expansion and splenic stem cell accumulation. Thus, tonic IFNs regulate gut microbial ecology, which is crucial for maintaining physiologic immune homeostasis and preventing inflammation.

*For correspondence:
Isabelle.Marie@nyulangone.org
(IJM);
david.levy@med.nyu.edu (DEL)

## Introduction

Interferons (IFNs) are important mediators of innate and adaptive immunity. The generic term IFN comprises three types of cytokines: type I family (IFN-I) encoded by multiple genes, including primarily numerous IFN-α subtypes and IFN-β; type II family (IFN-II), with IFN-γ being its sole member; and type III (IFN-III) family consisting of several IFN-λ s. Each IFN family signals through a distinct heterodimeric cell surface receptor. All members of IFN-I bind a receptor termed IFNAR, which triggers activation of the Jak kinases Jak1 and Tyk2 that mediate tyrosine phosphorylation of two members of the signal transducer and activator of transcription (STAT) family, STAT1 and STAT2. Activated STAT1 and STAT2, along with interferon regulatory factor (IRF) 9, form the heterotrimeric complex ISGF3 that binds to interferon stimulated response elements in the promoters of hundreds of interferon stimulated genes (*Au-Yeung et al., 2013*). While IFN-III binds a distinct receptor composed of IL28Ra and IL10Rb subunits, this signaling cascade also activates ISGF3 and is largely overlapping with the pathway downstream of IFN-I (*Lazear et al., 2019*). In contrast, IFN-II, after binding its cognate receptor (IFNGR), signals predominantly through homodimers of STAT1 and stimulates a set of genes containing a gamma-activating sequence (GAS) (*Ivashkiv, 2018*). All these pathways converge on STAT1, and STAT1 deficiency or hypofunction leads to insensitivity to all types of IFN. As expected, human STAT1 deficiency results in increased susceptibility to both viral and mycobacterial

infections (*Dupuis et al., 2001*; *Dupuis et al., 2003*), and hematopoietic cell transplantation remains the only curative treatment (*Bustamante et al., 2014*). Inability of STAT1-deficient patients to thrive in the absence of mounting appropriate innate immune responses to microbes precludes study of a contribution of STAT1 in homeostasis. However, some STAT1 partial loss of function (LOF) patients suffer from chronic colitis, as well as severe infections (*Thoeni et al., 2015*; *Sharfe et al., 2014*). On the other hand, individuals with STAT1 gain-of-function (GOF) mutations suffer most frequently from mucocutaneous diseases, in part due to depressed levels of $T_H17$ cells, thereby attributing important regulatory functions to STAT1 (*Liu et al., 2011*; *Puel, 2020*).

Despite engaging similar downstream signaling cascades, it is becoming increasingly evident that IFN-I and -III play distinct roles in establishing innate immunity against microbes and participate differently in the overall immune functions of the host (*Broggi et al., 2019*). For instance, IFN-III has been shown to resolve inflammation by reducing the number of IL-17-producing $T_H17$ helper T cells and restricting the recruitment of neutrophils (*Blazek et al., 2015*). Interestingly, besides its important role in inflammation, IL-17 also plays a significant role in hematopoiesis (*Krstic et al., 2012*). For instance, IL-17 stimulates myeloid and erythroid progenitors (*Lubberts et al., 2001*; *Krstic et al., 2010*), suggesting that IFN-III could play a role in the regulation of hematopoiesis through limiting the action of IL-17. Moreover, in part because of the more limited distribution of their receptor, IFN-III members display a unique capacity to counter pathogen invasion at mucosal sites while curbing overexuberant inflammation that helps maintain barrier integrity (*Pott and Stockinger, 2017*).

There is mounting evidence that commensal gut microbiota that exist at mucosal surfaces also play fundamental roles in shaping the host immune system (*Belkaid and Hand, 2014*; *Gutierrez-Merino et al., 2020*; *Castillo-Álvarez et al., 2018*; *Alteber et al., 2018*; *Canesso et al., 2018*). However, there is only limited understanding of the importance of the interplay between the microbiome and IFN to prevent pathogenesis. Herein, we unravel a role for STAT1 in the control of microbiota ecology that prevents inflammation and maintains immune homeostasis in the absence of an infectious challenge.

## Results

### STAT1-deficient mice exhibit splenomegaly, neutrophilia, and increased splenic progenitors

We observed that STAT1-deficient animals developed a spontaneous splenomegaly irrespective of age and sex. STAT1 KO spleens were found to average 5–10 times larger than their wild-type (WT) littermates (*Figure 1A*) in absence of any notable infection. Moreover, the normal splenic architecture of STAT1 KO animals was disrupted, with prominent signs of extramedullary hematopoiesis (*Figure 1A*). The number of total white blood cells (WBC, p=0.000007), as well as neutrophil (p=0.000034) and monocyte subsets (p=0.000029), was dramatically increased in STAT1 KO mice compared to WT (*Figure 1B,C*).

To better characterize the profound alterations of the STAT1 KO hematopoietic system, we undertook a more comprehensive analysis of blood populations using multiparameter flow cytometry (*Supplementary file 1*, *Figure 1—figure supplement 1*). WT and STAT1 KO blood were compared by visualization of Uniform Manifold Approximation and Projection (UMAP) plots. Profound changes in the distribution of the major classes of leukocytes were observed as shown in *Figure 1C*. Consistent with the analysis of blood counts, a dramatic elevation in the percentage of monocytes and neutrophils was noted. The percentage of neutrophils increased from an average of 7 % in WT animals to about 50 % of total blood cells in STAT1-deficient mice (*Figure 1D*). Conversely, STAT1 KO blood showed a reduced percentage of B lymphocytes and NK cells when compared to WT littermates. Interestingly, although the percentage of T cells did not change, a notable shift from naïve and central memory to effector T cells was observed for both $CD4^+$ and $CD8^+$ populations (*Figure 1D*). The same qualitative changes were observed when the analysis was performed on spleens, except that the percentage increase of effector T cells was less pronounced (*Figure 1E*). Notably, because of the greatly increased cellularity of STAT1 KO spleens, the total number of monocytes and neutrophils was approximately 100-fold higher in the KO spleen compared to WT (*Figure 1F*). Likewise, despite the differences in the percentage of effector T cells not being statistically significant (p=0.28 for CD4; p=0.36 for CD8), STAT1 KO spleens contained close to 10 times more $CD4^+$ and $CD8^+$ effector T cells

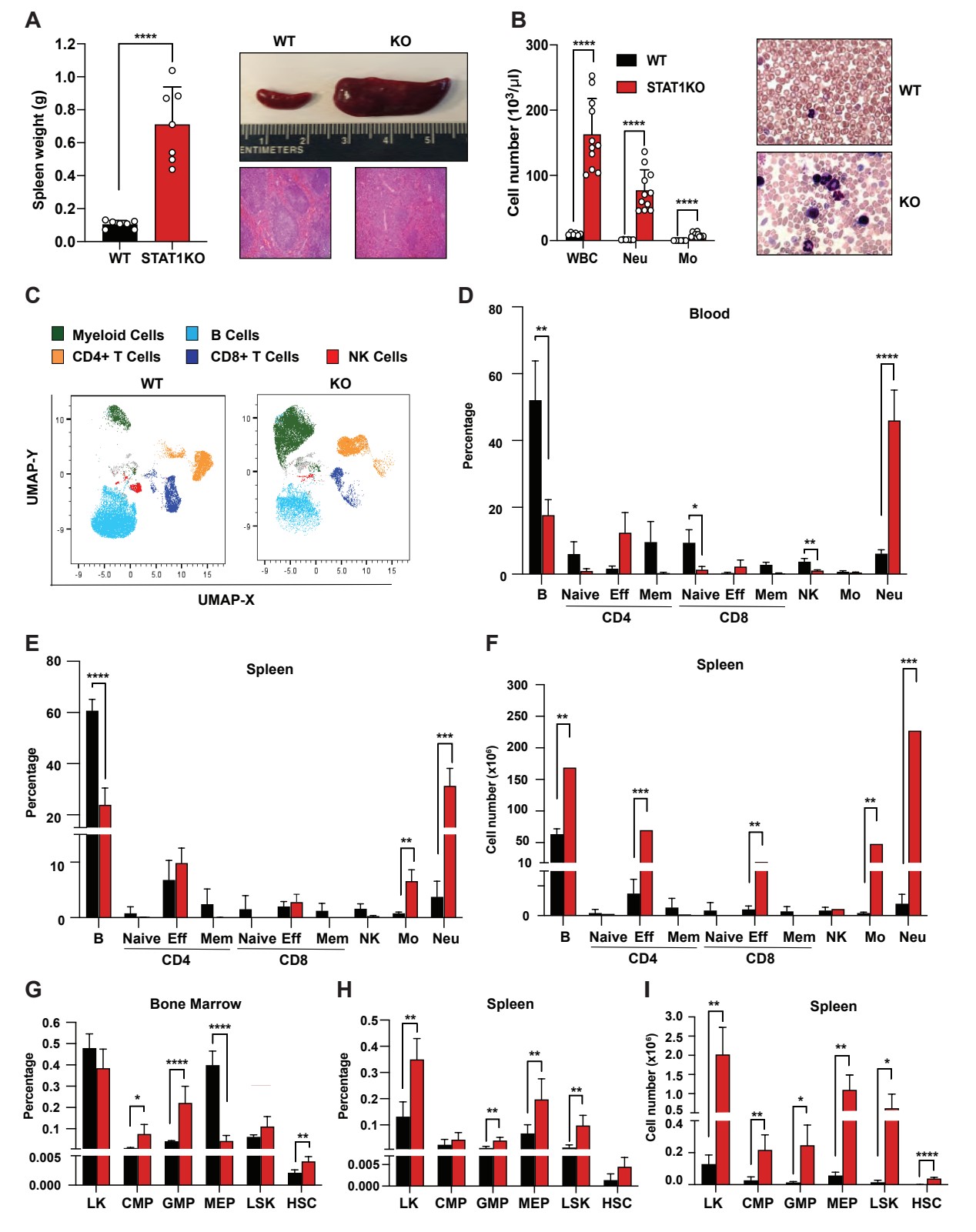

**Figure 1.** Characterization of STAT1 KO hematopoietic defects. (**A**) Spleen weights in g (n = 7), mean ± SD (left). Representative spleen of a STAT1 KO mouse and its age- and sex-matched WT littermate (top right). Representative histology of STAT1 KO and WT spleens, stained with H&E (bottom right). (**B**) Total WBC, neutrophil and monocyte counts (n = 6 WT, n = 11 KO), mean ± SD (left). Representative blood smear from WT and STAT1 KO blood after Giemsa staining (right). (**C**) Uniform Manifold Approximation and Projection (UMAP) plots of blood from WT and STAT1 KO littermates after

*Figure 1 continued on next page*

Figure 1 continued

18-color multi-parameter flow cytometry staining. Clusters were annotated using known markers. Flow cytometric analysis of blood (**D**), bone marrow (**G**), spleen (**E–H**) of STAT1 KO, and WT littermates (n = 4–5). Values represent mean ± SD of live cells, percentage or number, as indicated. *p<0.05, **p<0.01, ***p<0.001, ****p<0.0001 by Student's t-test. NK, natural killer; LK, lin⁻Sca1⁻c-Kit⁺; CMP, common myeloid progenitor; GMP, granulocyte-macrophage progenitor; MEP, megakaryocyte-erythroid progenitor; LSK, lin⁻Sca1⁺c-Kit⁺; HSC, hematopoietic stem cells. Each dot represents an individual animal (**A**, **B**).

The online version of this article includes the following figure supplement(s) for figure 1:

**Figure supplement 1.** Gating strategies of multi-parameter flow cytometry analysis.

**Source data 1.** Source data for *Figure 1A, B, D–I* in Excel format.

as compared to WT spleens, and these quantitative differences were highly statistically significant (*Figure 1E, F*, p=0.00023 for CD4; p=0.0058 for CD8).

The dramatic changes observed in the populations of mature cells in blood and spleen prompted us to examine stem and progenitor cells in bone marrow and spleen. Analysis of progenitor populations in the bone marrow showed altered progenitor population ratios, with a significant increased percentage of GMP (p=0.0009) and decreased percentage of MEP in STAT1 KO compared to WT (*Figure 1G*). In contrast, we observed an increased percentage and number of MEP in the spleens of STAT1 KO mice, consistent with the presence of extramedullary hematopoiesis. More strikingly, phenotypic hematopoietic stem cells (HSCs), defined as Lin⁻Sca1⁺c-Kit⁺CD150$^{hi}$CD48$^{low}$, which are very low in WT spleen, were significantly expanded in STAT1-deficient spleens (*Figure 1H,I*, p=0.00006).

## Definitive HSCs populate STAT1 KO spleens

In order to confirm that the Lin⁻Sca⁺Kit⁺CD150$^{hi}$CD48$^{low}$ cells present in STAT1 KO spleens were functional HSCs, we first tested their self-renewing potential in vitro. We compared the plating efficiency in myeloid promoting methylcellulose cultures of splenic and bone marrow cells from WT and STAT1 KO animals. Plating efficiency of STAT1-deficient bone marrow cells was higher than WT for the first plating, but subsequent passages showed only minor differences between the two genotypes (*Figure 2A*). In contrast, STAT1-deficient splenic progenitors showed much higher plating efficiency compared to WT, as well as greatly enhanced replating capacity. Indeed, STAT1-deficient splenic progenitors still gave rise to a significant number of colonies after four sequential passages in methylcellulose (*Figure 2B*). This result provides evidence that STAT1-deficient spleens host a large population of self-renewing progenitors that is not normally found in WT animals.

Since self-renewal of hematopoietic cells in vitro largely scores the presence of short-term progenitors, we wanted to assess the presence of functional long-term (LT) HSCs in STAT1-deficient spleens. Therefore, we performed a transplantation experiment into lethally irradiated recipient animals (CD45.1) using both spleen and bone marrow of WT and STAT1 KO animals (CD45.2) as donor cells. Reconstitution of the hematopoietic system by donor cells was monitored after 1 month. WT splenic cells were not capable of thriving in irradiated hosts, with the few transplanted animals that survived showing the absence of donor-derived cells (*Figure 2C*), indicative of the limited presence of HSCs in this organ under normal conditions (*Morita et al., 2011*). In contrast, STAT1-deficient splenic cells reconstituted the host hematopoietic system as efficiently as WT or STAT1 KO BM, with a median reconstitution of over 80 % donor-derived cells (*Figure 2C*). We next harvested the bone marrow and spleen from primary engrafted recipients and transplanted these cells into irradiated secondary recipients (CD45.1), in order to distinguish between short-term and long-term engraftment capability. Results from the secondary transplantation experiments showed even greater differences between WT and KO cells. Neither bone marrow nor spleen cells isolated from primary reconstituted animals that had received STAT1 KO bone marrow were capable of complementing secondary transplanted recipients, suggesting that STAT1 KO bone marrow was deficient in LT-HSCs (*Figure 2D*). In marked contrast, bone marrow cells from primary animals reconstituted with STAT1 KO spleen cells had a reconstituting capacity indistinguishable (p=0.43) from animals initially reconstituted with WT bone marrow (*Figure 2D*). Finally, in order to confirm the pluripotency of the progenitors found in STAT1-deficient spleens, blood from secondary transplanted recipients was analyzed after 4 months. We found no significant differences of the lineage allocation of blood amongst secondary recipients (*Figure 2E*). These results strongly suggest that fully functional pluripotent LT-HSCs reside in STAT1-deficient spleens, that these cells, albeit residing in the spleen of STAT1-deficient animals, home to

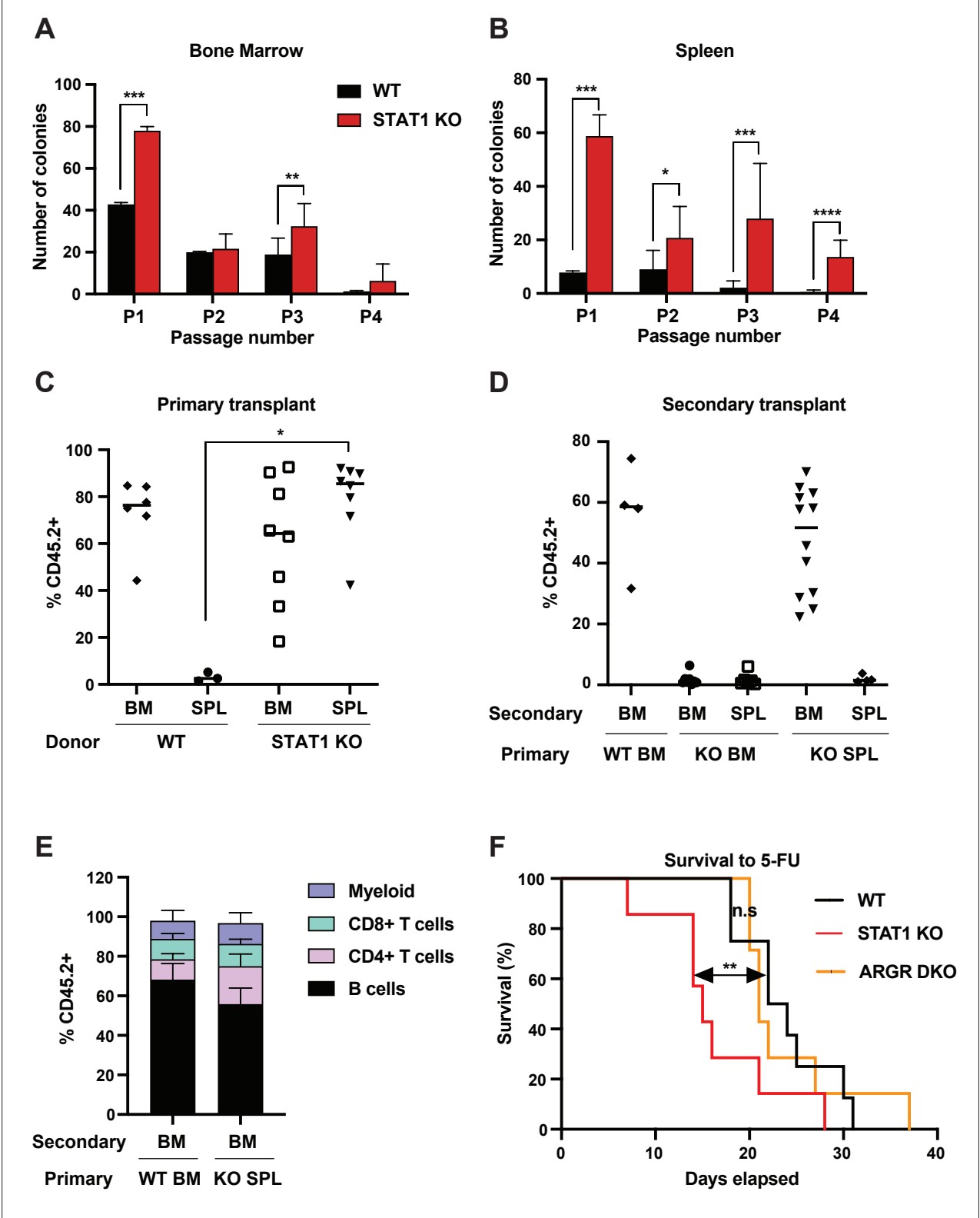

**Figure 2.** STAT1-deficient spleens harbor a large number of definitive HSCs. In vitro self-renewal colony forming assay of hematopoietic progenitors from bone marrow (**A**) or spleen (**B**) after serial plating (n = 3 mice, each plating in triplicate). Values represent mean number of colonies ± SD. Each replating culture was inoculated with equal numbers of cells. Quantification of donor-derived (CD45.2) cells, provenance as indicated, in the peripheral blood of recipient CD45.1 animals 1 month after primary transplant (**C**) or 4 months after secondary transplant (**D**). Values represent individual animals.

*Figure 2 continued on next page*

*Figure 2 continued*

(**E**) Quantification of donor-derived (CD45.2) B (B220⁺) cells, T (CD4⁺ or CD8⁺), and myeloid (CD11b⁺) cells in the peripheral blood of recipient animals 4 months after secondary transplant ; values represent mean ± SD; WT n = 4 mice; KO n = 11 mice. (**F**) Kaplan–Meier survival curve of WT, STAT1 KO, and ARGR DKO mice after weekly injections of 5-fluorouracil (5-FU). WT n = 5 mice, STAT1 KO and ARGR DKO n = 7 mice. *p<0.05, **p<0.01, ***p<0.001, ****p<0.0001 by Student's t-test or Gehan–Breslow–Wilcoxon test for Kaplan–Meier.

The online version of this article includes the following figure supplement(s) for figure 2:

**Source data 1.** Source data for *Figure 2A–F* in Excel format.

the BM following engraftment of WT animals, and that STAT1 KO BM is either depleted of LT-HSCs or these cells are functionally deficient. Of note, no myeloid expansion or splenomegaly was detected in animals transplanted with STAT1 KO bone marrow or spleen, suggesting that loss of STAT1 solely in hematopoietic cells is not sufficient to cause the myeloproliferative disease observed in STAT1 KO mice.

We considered whether the increased abundance of splenic LT-HSCs present in STAT1 KO animals represented an altered proliferative capacity. To investigate this possibility, we tested the sensitivity of STAT1 KO mice to 5-fluorouracil (5-FU), a myelosuppressive chemotherapeutic that kills proliferating cells. Following weekly injections of 5-FU, we documented that STAT1 KO mice died significantly faster than WT (p=0.01), and also faster than mice lacking both IFNAR and IFNGR (ARGR DKO) (*Figure 2F*). Since the lethality of 5-FU is related to depletion of cycling stem and progenitor cells, this result suggests that the abundant HSCs observed in STAT1 KO spleen are more sensitive to depletion than WT stem cells, perhaps due to decreased quiescence or more rapid exhaustion following cycling. Increased progenitor proliferation would be consistent with the enhanced replating efficiency of these cells observed in vitro (*Figure 2B*). Notably, the response of ARGR DKO animals to 5-FU challenge was indistinguishable (p=0.73) from WT animals, suggesting that IFN-I and IFN-II are not required to regulate HSC quiescence, at least in the presence of IFN-III.

## STAT1-deficient CD4 T cells produce elevated levels of IFN-γ and IL-17A

In order to better understand the underlying mechanisms prompting the inflammatory phenotype observed in a STAT1-deficient hematopoietic system, we undertook a comprehensive cytokine and chemokine profiling of the sera of these animals (Data *Supplementary file 1*). IFN-γ was increased approximately 300-fold in the sera of STAT1 KO animals and was moderately elevated in ARGR DKO mice (*Figure 3A*; p<0.0001). IL-5 and TNF-α showed a similar profile but reduced magnitude relative to IFN-γ (*Figure 3A*). However, a more striking difference was noted for IL-17A, which was found to average 100-fold higher concentration in STAT1 KO sera (p=0.04), with little or no elevation in ARGR DKO (*Figure 3A*). Since ARGR DKO animals, despite their lack of responsiveness to IFN-I and -II, showed little to no sign of pathology, the high concentrations of circulating IL-17A correlated with disease.

Given the observation of elevated cytokine levels, we sought to determine what cells were the source of IFN-γ and IL-17A. Therefore, we assayed peripheral blood lymphocytes by intracellular cytokine flow cytometry. We found that CD4⁺ T cells and NK cells were the main populations making IFN-γand IL-17A. As many as 30 % of STAT1-deficient CD4⁺ T cells produced IFN-γ (p=0.0061) as compared to only a few percent in WT controls (*Figure 3B and C*). Similarly, a larger percentage of STAT1 KO CD4⁺ T and NK cells made IL-17A when compared to WT counterparts (*Figure 3D*; p=0.0005). Moreover, not only was the number of IL-17 producing cells greater in STAT1 KO animals, but the amount of cytokine produced per cell was much greater, as indicated by a 5- to 10-fold increased mean fluorescence intensity (*Figure 3E*; p<0.0004).

## STAT1 KO pathology is associated with combined insensitivity to IFN-I, -II, and –III

STAT1 is a major transcription factor mediating the canonical JAK-STAT pathways downstream all three types of IFN. However, IFNs can also activate accessory pathways that could become preponderant in the absence of STAT1 and trigger an imbalance of the hematopoietic system homeostasis (*Gimeno et al., 2005*). To further evaluate a role for IFN, we applied a genetic approach and bred our STAT1 KO colony with mouse strains deficient for IFNAR and IFNGR that renders them insensitive

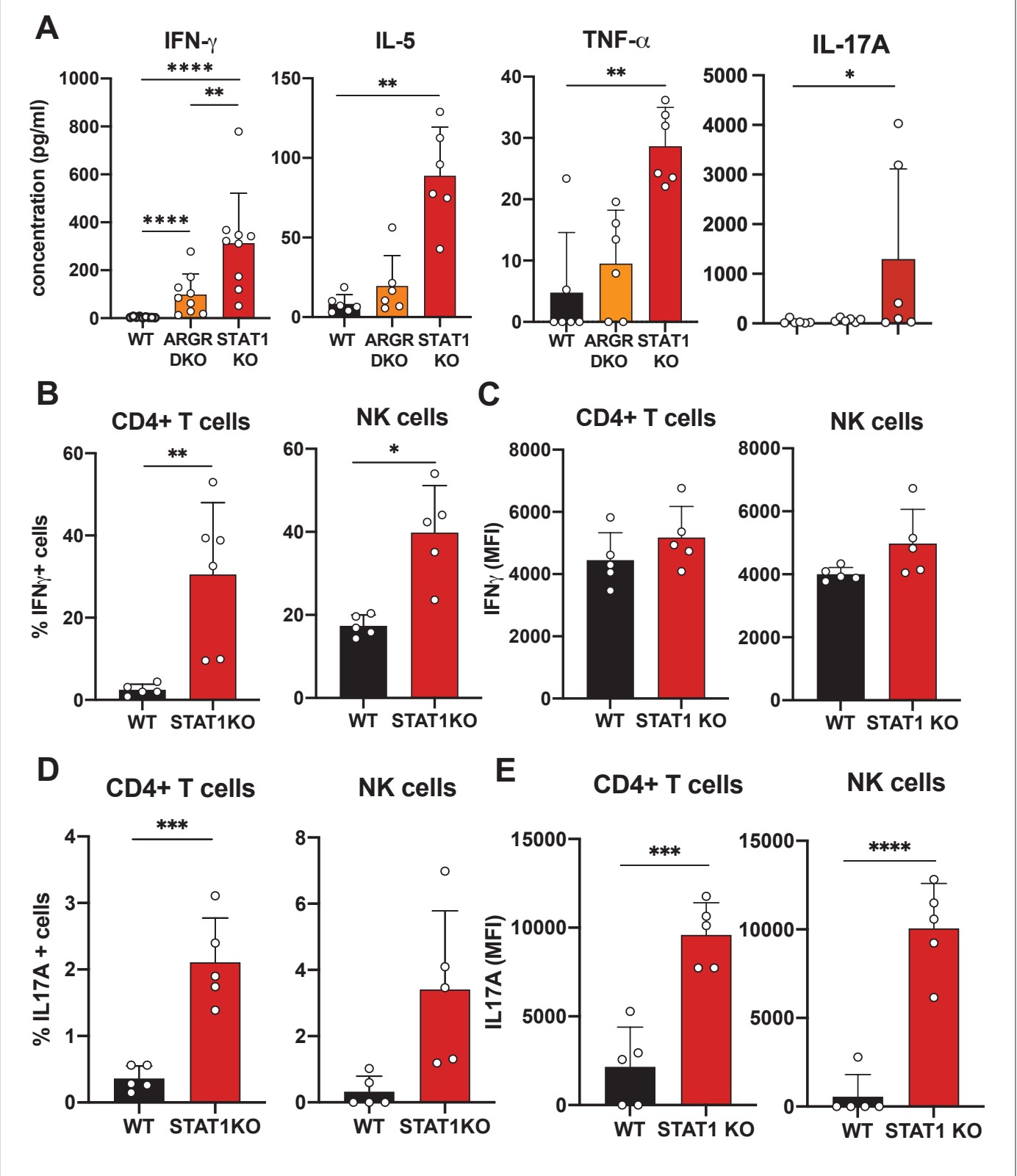

**Figure 3.** Cytokine profiling of STAT1 KO blood. (**A**) Cytokine concentration in blood of WT, ARGR DKO, and STAT1 KO mice in pg/ml (n = 6–9 mice for each group). Data are representative of two experiments. Percentage of IFN-γ (**B**) or IL-17A-producing (**D**) CD4+ T cells (left) or NK cells (right) in STAT1 KO and WT mice (n = 5 or 6) with corresponding MFI (**C, E**). Values represent mean ± S.D. *p<0.05, **p<0.01, ***p<0.001, ****p<0.0001 by Mann–Whitney (**A**) or Student's t-test (**B–E**). Each dot represents an individual animal.

*Figure 3 continued on next page*

*Figure 3 continued*

The online version of this article includes the following figure supplement(s) for figure 3:

**Source data 1.** Source data for *Figure 3A–E* in Excel format.

**Source data 2.** Comprehensive cytokine and chemokine quantification data related to *Figure 3* in Excel format.

to type I and II IFNs. The resulting triple deficient strain that lacked IFN-I and IFN-II receptors (ARGR STAT1 TKO) still displayed the same altered parameters that we observed in mice singly deficient for STAT1 (*Figure 4—figure supplement 1*). Therefore, alternate signals downstream of IFN receptors are unlikely culprits of the observed pathology. However, loss of IFN-I and II signaling (ARGR DKO) did not mimic the phenotype mediated by STAT1 loss. These data suggested that STAT1 was essential to immune fitness but the role of IFNs remained unclear.

To further investigate roles for IFNs upstream of STAT1 in providing protection from hematopoietic dysregulation, we examined the phenotypes of mice unable to respond to IFN-I, -II, and –III. As previously noted, ARGR DKO mice showed very mild signs of an altered immune system compared to STAT1 KO animals. Despite being unresponsive to IFN-I and -II, these animals did not present with splenomegaly and displayed very mild neutrophilia (*Figure 4A*). Interestingly, these animals did not exhibit an increase of CD4$^+$ effector T cells as observed in the STAT1 KO blood and spleen (*Figure 4B*). In contrast, STAT2 KO animals, which are impaired in their response to IFN-I and -III, showed profound defects that largely recapitulated what was observed in STAT1-deficient mice, except that no increase in the percentage of CD8+ effector T cells was detected in blood (*Figure 4B-D*). Taken together, these data suggested that IFN-I and -III are the main cytokines acting through STAT1 to provide a protective environment to insure proper homeostasis of the immune system, but additional STAT1-dependent functions are also at play. To better examine a role for type II IFN in this process, we compared STAT2 KO and IFNGR/STAT2 (GR STAT2) DKO mice. GR STAT2 DKO animals cannot respond to any type of IFN, similar to the IFN signaling defect in STAT1 KO mice. This strain also provided the advantage of probing any role for additional cytokines or growth factors that activated STAT1, but not STAT2. Detailed flow cytometry analysis of the blood and spleen of GR STAT2 DKO mice uncovered defects virtually identical to the ones observed for STAT1 KO mice (*Figure 4B and C*), strongly indicating that the phenotype of STAT1- deficient mice is associated with combined insensitivity to IFN-I, -II, and -III (*Supplementary file 2*).

Interestingly, analysis of bone marrow progenitors showed an increase of CMP and GMP accompanied by a decrease of MEP representation for all mutant genotypes studied (*Figure 4D*). More importantly, the number of splenic HSCs was dramatically increased in STAT1 KO (p=0.00007) and GR STAT2 DKO mice (p=0.026), and this increase was not observed in the other mutant strains (*Figure 4E*). Taken together, our results strongly suggested that the combined absence of IFN-γR and STAT2 (GR STAT2 DKO) phenocopied STAT1 deficiency. To confirm this hypothesis, we measured IFN-γ and IL-17A production by CD4$^+$ T cells and NK cells in these animals. Both cytokines were elevated compared to WT (*Figure 4F and G*), mimicking the phenotype of STAT1 KO cells (*Figure 3*).

## STAT1-deficient mice are prone to develop colitis

In addition to the hematopoietic defects documented above, STAT1 KO mice developed spontaneous colitis, frequently accompanied by rectal prolapse. Gross anatomy of the STAT1-deficient colons revealed increased angiogenesis (*Figure 5A*), a characteristic of inflamed bowels (*Alkim et al., 2015*). Histopathologic analysis of colon sections documented extensive im mune cell infiltrates, epithelial hyperplasia and goblet cell loss in STAT1 KO colons, associated in some cases with the presence of cryptitis (*Figure 5B,C*). Damage to the gut epithelium often results in alteration of the intestinal barrier integrity and bacterial leakage into neighboring organs. Therefore, we scored for presence of bacteria in the liver, mesenteric lymph nodes and spleens of STAT1 KO mice compared to WT animals. Bacteria were observed mainly in the liver in STAT1 KO mice and to a lesser extent in the mesenteric lymph nodes and very rarely in the spleen (*Figure 5D*), while organs from WT animals were largely sterile. In addition, we tested the sensitivity of STAT1 KO mice to dextran sulfate sodium (DSS)-induced colitis. Signs of DSS-induced intestinal inflammation, as evidenced by colon shortening and altered histopathology, were more pronounced in STAT1-deficient mice than WT littermates (*Figure 5E,F*).

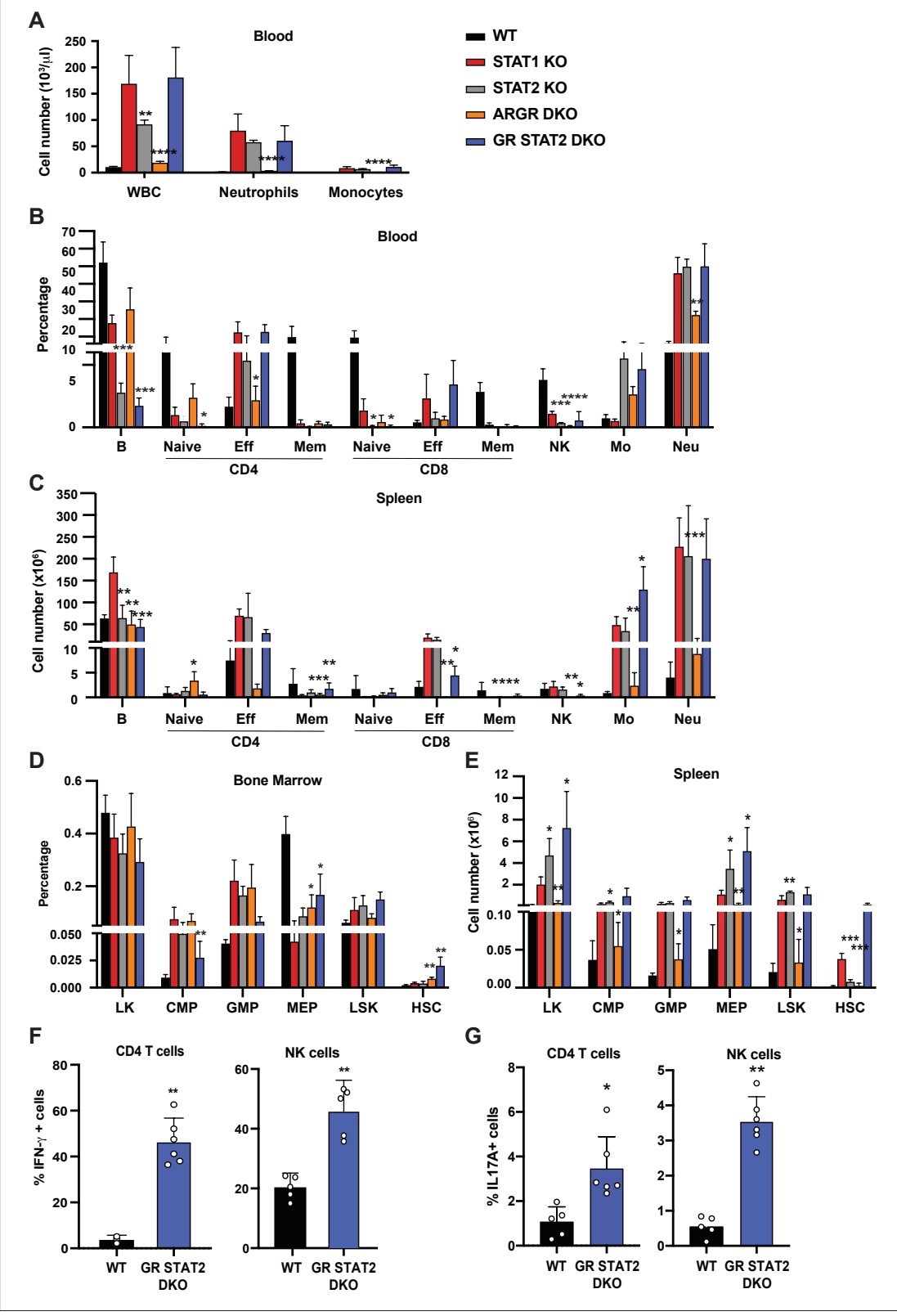

**Figure 4.** Combined IFN type I, II, III response deficiency recapitulates STAT1 deficiency. (**A**) Total WBC, neutrophils, and monocytes counts using Hemavet in blood from WT, STAT1 KO, STAT2 KO, ARGR DKO, and GR STAT2 DKO mice (n = 5–10 mice/group). Quantification by flow cytometry of mature cell populations from blood (**B**), spleen (**C**), and progenitor populations from bone marrow (**D**) and spleen (**E**) from WT, STAT1 KO, STAT2 KO, ARGR DKO, and GRSTAT2 DKO (n = 4 or 5). Same legend as in (**A**). Statistical comparison is between each genotype and STAT1 KO. Percentage

*Figure 4 continued on next page*

*Figure 4 continued*

of IFN-γ- (**F**) or IL-17A-producing (**G**) CD4+ T cells (left) or NK cells (right) in GR STAT2 DKO and WT mice (n = 4–6). Values represent mean ± SD of live cells, percentage or number, as indicated. *p<0.05, **p<0.01, ***p<0.001, ****p<0.0001 by Student's t-test. NK, natural killer; LK, lin⁻Sca1⁻c-Kit⁺; CMP, common myeloid progenitor; GMP, granulocyte-macrophage progenitor; MEP, megakaryocyte-erythroid progenitor; LSK, lin⁻Sca1⁺c-Kit⁺; HSC, hematopoietic stem cells. Each dot represents an individual animal (**F, G**).

The online version of this article includes the following figure supplement(s) for figure 4:

**Figure supplement 1.** Flow cytometric analysis of myeloid cells in WT, STAT1 KO, ARGR DKO, and ARGR STAT1 TKO animals.

**Source data 1.** Source data for **Figure 4A–G** in Excel format.

## STAT1-deficient mice present with altered gut microbiota

Inflammatory bowel diseases (IBD) such as colitis are often associated with altered microbiota. Therefore, we investigated whether STAT1 deficiency could perturb the representation of gut microflora and cause dysbiosis. To this end, we performed 16 S rRNA gene sequencing on fecal DNA samples collected from age- and sex-matched animals from four different colonies: WT, STAT1 KO, ARGR DKO, and GR STAT2 DKO. Analysis of alpha diversity measured by the Simpson and Shannon diversity indexes revealed a significant reduction of bacterial taxa diversity in STAT1 KO and GR STAT2 DKO animals compared to WT, whereas the ARGR DKO microbiome was found to be as diverse as the controls (*Figure 6A and B*). Thus, commensal bacterial diversity was inversely correlated with the severity of the inflammatory disease. Furthermore, beta diversity represented by principal coordinate analysis (PCoA) underscored similarities in bacterial diversity between the different groups of animals partially or completely lacking IFN response (*Figure 6—figure supplement 1A*). Accordingly, analysis of the relative frequency of major bacterial families unveiled several significant differences between WT and mutant strains (*Figure 6C*), among them a large decrease of *Bacteroidales S24-7* and *Burkholderiales Alcaligenaceae* and a consistent increase of *Deferribacterales Deferribacteraceae* and *Campylobacterales Helicobacteraceae*, suggesting that the abundance of these species is controlled by IFN (*Figure 6D*). However, another family, *Bacteroidales Prevotellaceae*, was of particular interest, since increases were present only in STAT1 KO and GR STAT2 DKO animals, therefore correlating with disease severity (*Figure 6E*). Two predominant clusters of amplicon sequence variants (ASV) related to the *Prevotella* family were increased in diseased animals. Since these ASV had not been taxonomically assigned to any previously identified species, we generated a phylogenetic tree based on the 16 S rRNA sequences of these ASV and matched with known *Prevotella* taxa assignments (*Figure 6—figure supplement 1B*). This analysis revealed that these ASV clustered with *Prevotella heparinolytica*, a microbial species known to trigger $T_H17$ responses (*Calcinotto et al., 2018*).

To probe for a possible causative connection between the observed dysbiosis of STAT1 KO animals and disease, we treated STAT1 KO mice with a mixture of broad-spectrum antibiotics (ABX). Following 4 wk treatment, analysis of fecal DNA samples from ABX-treated mice showed a greater than 100-fold reduction in total bacterial DNA, consistent with an expected depletion of commensal microbiota. Strikingly, ABX-treated STAT1 KO mice did not display splenomegaly (*Figure 6F*; p=0.00011). Moreover, ABX treatment restricted the accumulation of splenic CMP, GMP, and MEP progenitors as well as HSCs in STAT1 KO animals (*Figure 6G*). Finally, peripheral blood leukocytes from ABX-treated and control STAT1 KO mice were examined for IL-17A production. The elevated IL-17⁺ CD4⁺ T cells characteristic of STAT1 KO mice were significantly decreased by ABX treatment (*Figure 6H*; p=0.0032). Interestingly, reduction of microbial load through broad-spectrum antibiotic treatment showed a similar effect on GR STAT2 DKO mice. ABX-treated GR STAT2 DKO mice did not display splenomegaly, increased representation of myeloid cells in blood or accumulation of splenic progenitors (*Figure 6—figure supplement 2*).

## IL-17Ra deficiency rescues splenomegaly and neutrophilia observed in STAT1-deficient mice

Inflammation in STAT1 KO mice strongly correlated with IL17 production and altered peripheral T cell profiles. To examine mechanisms underlying the T cell alterations, we first probed the ability of STAT1 KO CD4⁺ T cells to differentiate into $T_H17$ cells in vitro. No difference in the generation of $T_H17$ cells in vitro was observed between STAT1 KO and WT controls in response to IL-6 and TGF-β (*Figure 7—figure supplement 1A*). We next analyzed the capacity of STAT1 KO CD4⁺ T cells to differentiate

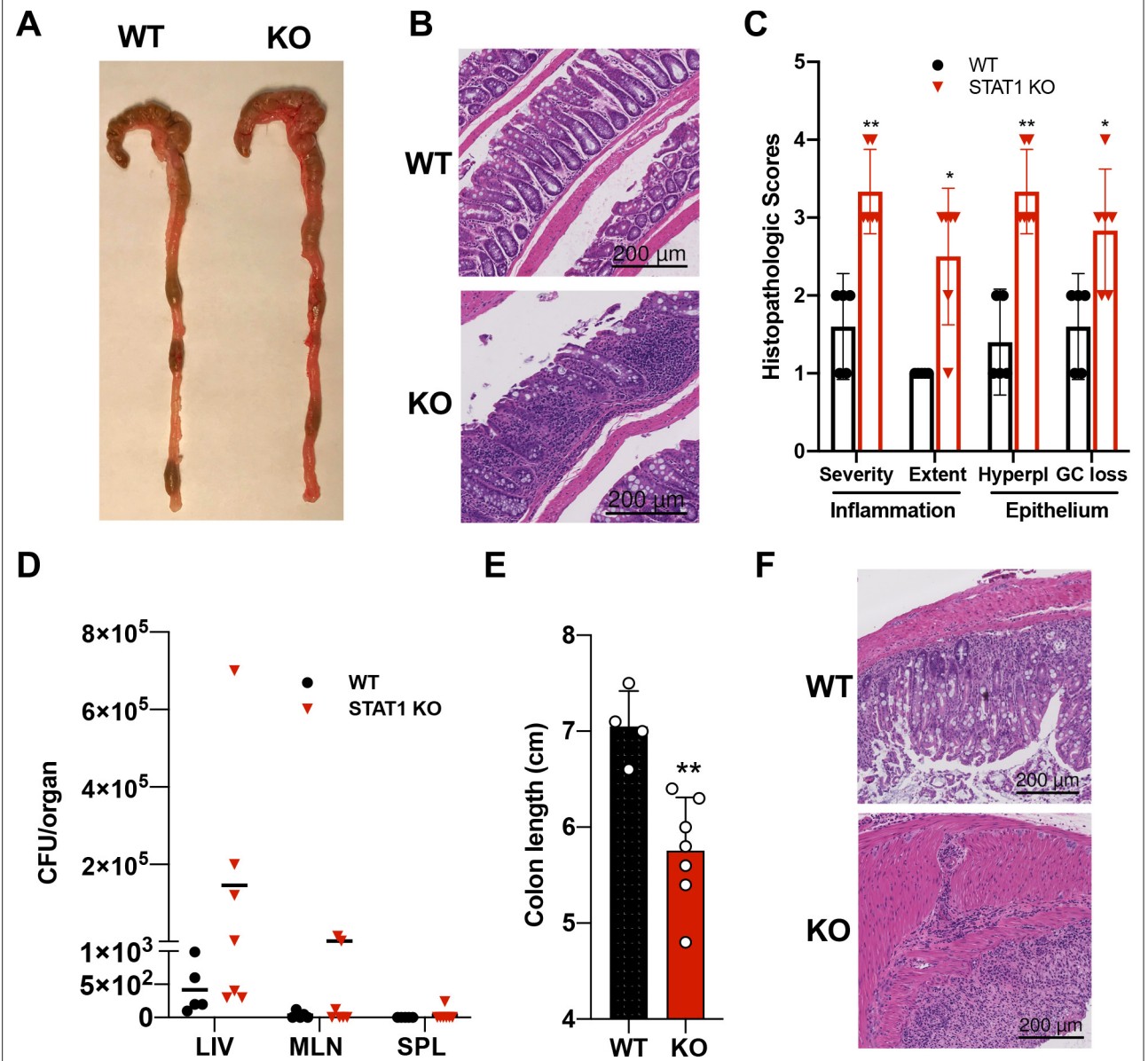

**Figure 5.** STAT1KO mice are prone to develop spontaneous colitis. (**A**) Representative colon gross anatomy of WT and STAT1 KO mice. (**B**) Representative histology of WT and STAT1 KO colons, stained with H&E (scale bar = 200 μm). (**C**) Histopathologic scores of WT and STAT1 KO colons. Scoring was performed for severity and extent of inflammatory cell infiltrates and for epithelial changes (hyperplasia [hyperpl] and goblet cell loss [GC loss]) (n = 5 for WT, n = 6 for STAT1 KO) *p<0.05, **p<0.01 by Mann–Whitney U test. (**D**) Bacterial leakage (CFU/organ) into liver (LIV), mesenteric lymph nodes (MLN) and spleen (SPL), as described in materials and methods (n = 5 for WT, n = 7 for STAT1 KO). (**E**) Colon length after 7 days of DSS treatment in drinking water (n = 4 for WT, n = 7 for STAT1 KO) **p<0.01 by Student's t-test. (**F**) Representative histology of WT and STAT1 KO colons, stained with H&E, after 7 days of DSS treatment (scale bar = 200 μm). Each symbol represents an individual animal (**C–E**).

into T$_{reg}$ cells in vitro in response to TGF-β and IL-2. Again, no significant deficit was observed in production of STAT1 KO T$_{reg}$ compared to WT (*Figure 7—figure supplement 1B*). Likewise, analysis of T cell populations in the thymus uncovered no statistically significant difference in thymic T$_{reg}$ populations (*Figure 7—figure supplement 1C*, p=0.6 by Mann–Whitney). However, contrasting with what we observed in vitro and in the thymus, peripheral T$_{reg}$ were virtually absent in STAT1 KO spleens (*Figure 7A*), consistent with colitis (*Yamada et al., 2016*) and with an inability to suppress T$_H$17 expansion.

To directly test whether IL-17 could drive the anomalies observed in STAT1-deficient animals, we generated a mouse strain deficient for both *Stat1* and *Il17ra* (STAT1 IL-17R DKO). Strikingly, blocking

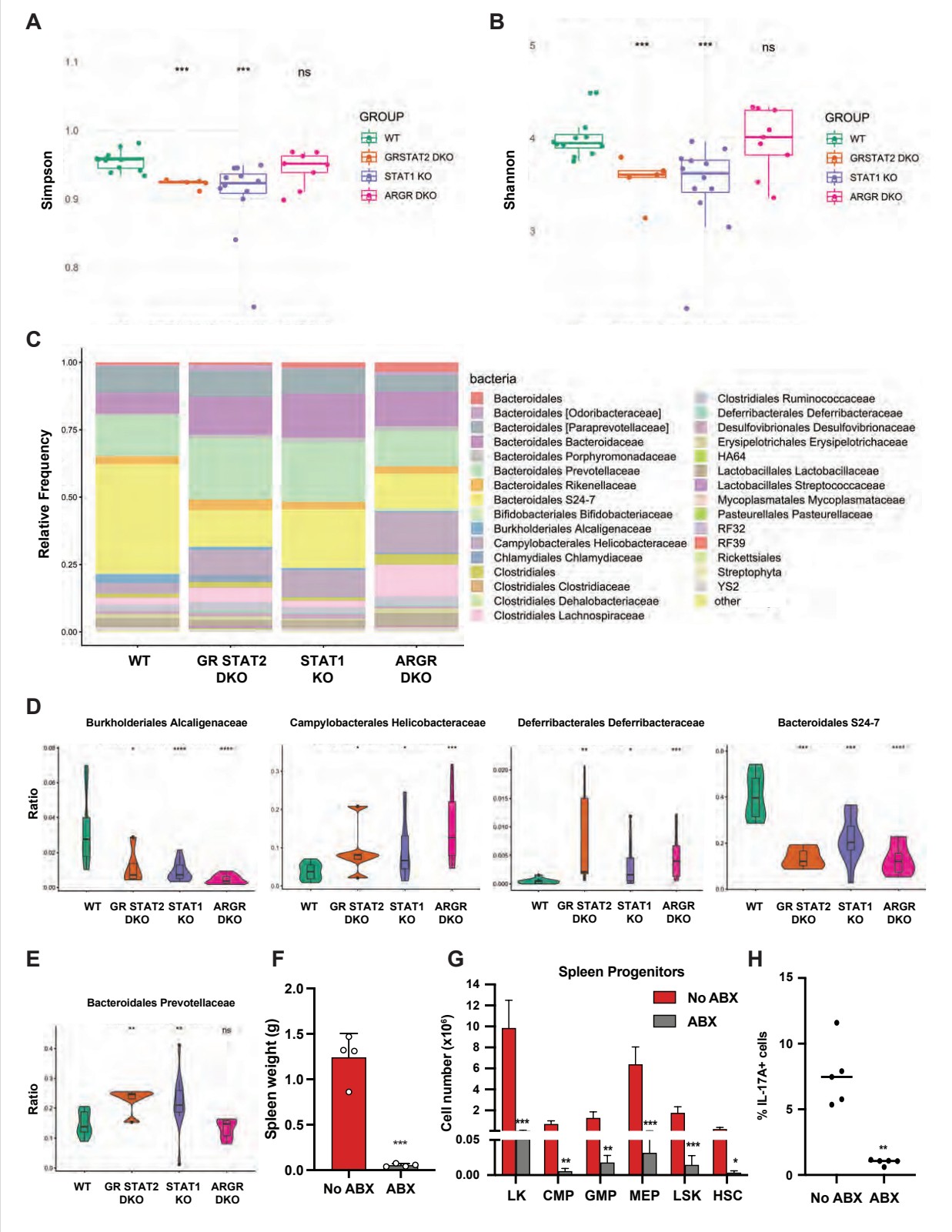

**Figure 6.** Tonic IFN controls gut microbiota 16 S rRNA sequencing of fecal DNA samples from WT, GR STAT2 DKO, STAT1 KO, and ARGR DKO mice (n = 8–12 animals/strain). (**A**) Alpha diversity as estimated by Simpson index (Kruskal–Wallis p-value = 0.00068). (**B**) Alpha diversity as estimated by Shannon index (Kruskal–Wallis p-value = 0.00082). (**C**) Family level microbiota composition. (**D**) Relative abundance of the total microbial load of four families of bacteria. (**E**) Relative abundance of the total microbial load of Bacteroidales Prevotellaceae. (**F**) Spleen weights in g of antibiotics (ABX)-treated

*Figure 6 continued on next page*

*Figure 6 continued*

STAT1 KO animals compared to untreated animals (No ABX) (n = 4). (**G**) Flow cytometric analysis of splenic progenitors of antibiotics (ABX)-treated STAT1 KO animals compared to untreated animals (No ABX) (n = 4). (**H**) Percentage of IL-17A-producing CD4+ T cells in antibiotics (ABX)-treated animals compared to untreated animals (No ABX). Values represent mean ± SD of live cells, percentage or number, as indicated. *p<0.05, **p<0.01, ***p<0.001, ****p<0.0001 by Student's t-test. LK, lin-Sca1-c-Kit+; CMP, common myeloid progenitor; GMP, granulocyte-macrophage progenitor; MEP, megakaryocyte-erythroid progenitor; LSK, lin-Sca1+ c-Kit+; HSC, hematopoietic stem cells. Each dot represents an individual animal (**A**, **B**, **F**, **H**).

The online version of this article includes the following figure supplement(s) for figure 6:

**Figure supplement 1.** Bioinformatic analysis of beta diversity and phylogeny of *Prevotella* species.

**Figure supplement 2.** Antibiotic treatment of GR STAT2 DKO mice.

IL-17 responses normalized the blood parameters of STAT1-deficient mice. As expected, IL-17Ra single deficient mice did not present any sign of perturbed hematopoiesis, as previously described (*Tan et al., 2006*; *Figure 7B*). An in-depth analysis of the representation of mature cells in the blood and spleen of these animals showed that STAT1 IL-17R DKO mice displayed at most a modest increase of the myeloid population (*Figure 7C and E*). Likewise, concomitant IL-17Ra deficiency lowered the percentage of GMPs observed in STAT1-deficient bone marrow and spleen, resolving this phenotype essentially to control levels (*Figure 7D and F*). More importantly, we observed no increase of splenic HSCs in STAT1 IL-17R DKO mice (*Figure 7F*).

In contrast, accumulation of IFN-γ and IL-17A-producing CD4$^+$ T cells and NK cells was not significantly different between STAT1 KO and STAT1 IL-17R DKO mice (*Figure 7—figure supplement 2A–D*). Accordingly, the percentage of circulating effector T cells was equivalent to that seen in STAT1 KO mice (*Figure 7C*). Analysis of plasma cytokine levels also showed increased abundance of inflammatory cytokines in STAT1 IL-17R DKO mice (*Figure 7—figure supplement 2E*). Interestingly, loss of IL-17 responsiveness reduced the abundance of IFN-γ, TNF-α, and IL-5, although all except IL-5 remained elevated compared to WT mice (p=0.003 for TNF-α; p=0.017 for IFN-γ). IL-17A appeared to be autoregulated, since its abundance was elevated in IL-17R KO mice (p=0.0022) and even higher in STAT1 IL-17R DKO mice (*Figure 7—figure supplement 2E*, p=0.0028). None of the other inflammatory cytokines tested was elevated in IL17R KO mice.

That effector T lymphocyte and cytokine levels remained elevated in STAT1 IL-17R DKO mice, in spite of the resolution of the myeloproliferative phenotype, suggests that they are primary responses to the lack of STAT1 signaling rather than to elevated IL-17A. Because we hypothesized that microbial dysbiosis is the primary trigger of inflammation in STAT1 KO mice, we examined gut microbial diversity of STAT1 IL-17R DKO mice. Microbial dysbiosis was still present in STAT1 IL-17R DKO animals, similar to what was previously observed for STAT1 and GR STAT2 KO mice (*Figure 7—figure supplement 3A,B*). Taken together, these results suggest that increased IL-17A signaling is the primary driver of the myeloproliferative phenotype in STAT1 KO animals but is secondary to microbial dysbiosis, alterations in lymphocyte subsets, and other inflammatory cytokines.

## Discussion

Our data provide evidence that tonic signaling through the IFN pathway, acting through its common mediator STAT1, is an essential modulator of immune homeostasis, reducing the propensity for inflammation. This tonic IFN signaling appears to involve all of the three major IFN families, since loss of all three arms of the pathway was required for the full inflammatory phenotype. The absence of tonic IFN signaling allows development of a skewed microbiome that triggers an inflammatory response, likely due to an increased bias of T$_H$17 cells toward a pathogenic phenotype and impaired homing or survival of peripheral T$_{reg}$.

A striking attribute of the inflammatory phenotype that developed in the absence of STAT1 was a profound myeloid hyperplasia, accompanied by the accumulation of stem and progenitor cells in peripheral lymphoid organs. This phenotype was fully recapitulated in GR STAT2 DKO mice, demonstrating that it is caused by loss of IFN signaling, rather than by a non-canonical function of STAT1, and it was fully ameliorated by loss of IL17Ra, showing that IL17 signaling is epistatic to STAT1 for this phenotype. Myeloid hyperplasia in the absence of STAT1 is not unprecedented, since a distinct strain of STAT1 KO mice (Balb/c) housed in an independent animal facility was previously reported to develop a myeloid proliferative disease accompanied by progenitor expansion (*Porpaczy et al.,*

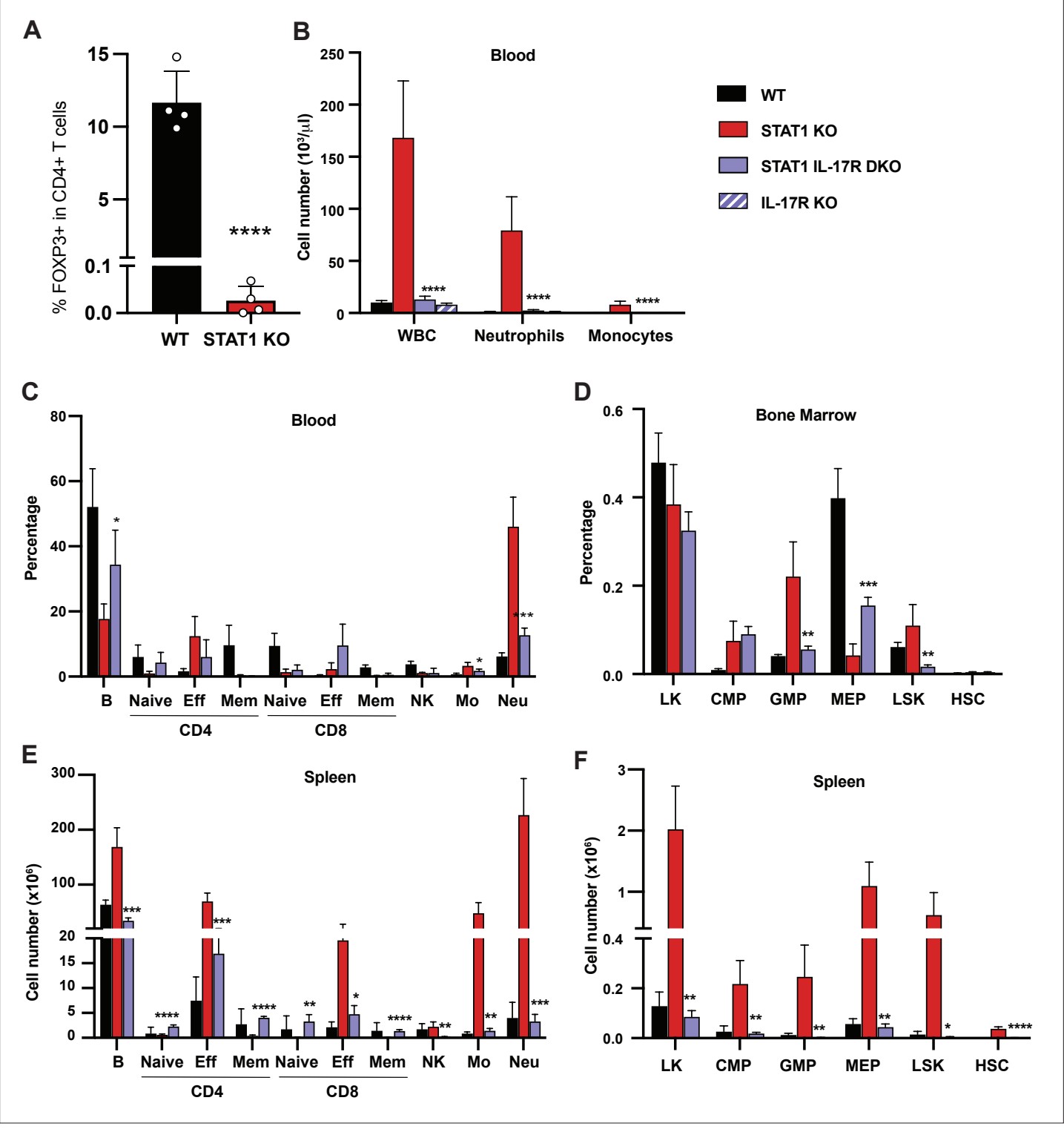

**Figure 7.** IL-17R deficiency rescues hematopoietic anomalies of STAT1 KO mice. (**A**) Percentage of FOXP3+ T_reg cells in splenic CD4+ T cells of WT and STAT1 KO littermates (n = 4). (**B**) Total WBC, neutrophil, and monocyte counts of WT, STAT1 KO, IL-17R STAT1 DKO, and IL-17R KO, mean ± SD (n = 6 for WT, n = 9 or 10 for other groups). Quantification by flow cytometry of mature cell populations from blood (**C**), spleen (**E**), and progenitor populations from bone marrow (**D**) and spleen (**F**) from WT, STAT1 KO, IL-17R STAT1 DKO (n = 4 or 5). Values represent mean ± SD of live cells, percentage or number, as indicated. *p<0.05, **p<0.01, ***p<0.001, ****p<0.0001 by Student's t-test. NK, natural killer; LK, lin⁻Sca1⁻ c-Kit⁺; CMP, common myeloid progenitor; GMP, granulocyte-macrophage progenitor; MEP, megakaryocyte-erythroid progenitor; LSK, lin⁻Sca1⁺c-Kit⁺; HSC, hematopoietic stem cells. Each dot represents an individual animal (**A**).

*Figure 7 continued on next page*

*Figure 7 continued*

The online version of this article includes the following source data and figure supplement(s) for figure 7:

**Figure supplement 1.** Analysis of STAT1 KO and WT T$_H$17 and Treg cells.

**Figure supplement 2.** Cytokine production by STAT1 IL-17R DKO lymphocytes.

**Source data 1.** Source data for *Figure 7A–F* in Excel format.

**Figure supplement 2—source data 1.** Comprehensive cytokine and chemokine quantification data related to *Figure 7—figure supplement 2* in Excel format.

**Figure supplement 3.** Microbiome diversity 16 S rRNA sequencing of fecal DNA samples from WT, GR STAT2 DKO, STAT1 KO, ARGR DKO, and IL17R STAT1 DKO mice (n = 5–21 animals/strain).

---

*2018*). Interestingly, this study also noted development of a B-cell malignancy in addition to myeloid expansion, evident following treatments to reduce the myeloid burden or after transplant. We did not observe B-cell malignancy in our STAT1 KO colony either before or after transplantation. However, Balb/c mice are known to be highly susceptible to B-cell malignancies (*Potter and Morrison, 1996*), a characteristic that has been mapped to the *Cdkn2a* locus, with Balb/c mice expressing an impaired version of the p16INK4a tumor suppressor (*Zhang et al., 1998*). The fact that an independent strain of mutant mice housed in a distinct facility developed the same myeloid proliferative disease reinforces the notion that loss of signaling through the STAT1 pathway is a fundamental cause.

Most of the inflammatory phenotype that develops in the absence of tonic IFN signaling is mediated by IL-17, as demonstrated by the normalization of the majority of hematopoietic parameters following deletion of *IL-17Ra*, consistent with current understanding. For instance, IL-17 has been shown to stimulate myeloid progenitors and early-stage erythroid progenitors. Overexpression or exogenous administration of IL-17 stimulates granulopoiesis and the recruitment of both short- and long-term stem cells (*Lubberts et al., 2001*), relocating core erythropoiesis from bone marrow to spleen (*Krstic et al., 2010*). Our data are also consistent with previous studies, including studies of human patients with mutations in *STAT1*, implying that STAT1-activating cytokines such as IFNs inhibit T$_H$17 differentiation (*Batten et al., 2006*; *Yoshimura et al., 2006*; *Stumhofer et al., 2006*; *Amadi-Obi et al., 2007*; *Feng et al., 2008*; *Kimura et al., 2008*; *Tanaka et al., 2008*; *Chen et al., 2009*; *Ramgolam et al., 2009*; *Diveu et al., 2009*; *El-behi et al., 2009*; *Villarino et al., 2010*; *Liu and Rohowsky-Kochan, 2011*; *Okada et al., 2020*). Much attention has also been paid to a role for IL-27, a cytokine capable of dramatically inhibiting T$_H$17 cell commitment in a STAT1-dependent manner (*Wang and Liu, 2016*). However, our data argue for a dominant role for IFNs over IL-27, since the proinflammatory phenotype due to STAT1 loss, including enhanced production of IL-17, was mimicked by the combined absence of STAT2 and IFNGR, thereby inactivating IFN-I/III and blocking IFNγ responses, but not impairing the action of IL-27, which does not rely on STAT2.

T$_H$17 cells differentiate as a spectrum of phenotypically similar cells from homeostatic protective functions to inflammatory pathogenic cells (*Yoshimura et al., 2006*; *Gaublomme et al., 2015*; *Ciofani et al., 2012*). Consistent with our data, it has been suggested that STAT1 acts at an early stage of T$_H$17 differentiation to impair pathogenesis (*Yosef et al., 2013*), although its molecular targets and the mechanisms regulating T$_H$17 pathogenic potential remain incompletely understood. Interestingly, analysis of a published T$_H$17 single-cell expression dataset (*Karmaus et al., 2019*) indicated that IFN-stimulated genes are largely restricted to non-pathogenic cells. A possible explanation for altered gene expression in mutant cells is that the absence of STAT1 strengthens signaling by other STATs, resulting in biased T$_H$ differentiation. Direct antagonistic actions between STAT1 and STAT3 (*Regis et al., 2008*; *Wan et al., 2015*) and between STAT1 and STAT4 (*Nguyen et al., 2002*) have been described. Over-activation of STAT3 or STAT4 due to absence of normal competition from STAT1 could partially underlie the increases in IL-17 and IFN-II observed in KO T cells. However, the fact that loss of STAT2 recapitulates most if not all of the symptoms of a STAT1 KO mouse would argue that such a competition mechanism is at most a minor contributor to the inflammatory process.

Loss of peripheral T$_{reg}$ cells was a striking phenotype of STAT1 KO mice. This loss did not appear to result from impaired T$_{reg}$ differentiation, since no defects were observed in vitro or in the thymus. Peripheral T$_{reg}$ loss could reflect a defect in homing to or survival in peripheral lymphoid organs and/or tissues. Decreased homing could be related to the drastic reduction of CD62L, a crucial lymphoid homing molecule (*Wirth et al., 2009*), observed on STAT1-deficient T cells (*Figure 1*). However, it is

quite likely that other cell type or the microbiota also regulate the T$_{reg}$ pool (*Atarashi et al., 2011*; *Furusawa et al., 2013*).

An important contributor to hematopoietic homeostasis is tonic IFN signaling (*Gough et al., 2012*). A role for the gut microbiome in regulating the response to systemic type I IFN has been documented (*Bradley et al., 2019*; *Mitchell et al., 2012*; *Ganal et al., 2012*; *Steed et al., 2017*; *Ito et al., 1976*). Recent studies have also substantiated the role of commensal microbiota as the major regulator of steady state IFN production (*Gutierrez-Merino et al., 2020*; *McAleer and Kolls, 2012*; *Abt et al., 2012*), for example by identifying microbiota-controlled tonic IFN expression by pDCs to be instrumental to 'instruct' responsiveness of cDCs, poising them for future immune intervention (*Schaupp et al., 2020*). The main pathways that have been implicated in triggering IFN production by the microbiota depend on Toll-like receptors (TLR) and downstream adaptors, such as Myd88 and TRIF (*Kawashima et al., 2013*), or through the recognition of microbial nucleic acids by the cGAS/STING and RLH pathways (*Roers et al., 2016*). However, it recently came to light that bacterial metabolites, such as secondary bile acids, impact IFN production and responses as well (*Steed et al., 2017*; *Chang et al., 2004*; *Grau et al., 2020*; *Winkler, 2020*).

Impaired regulation of T cells in the absence of IFN signaling contributes to the propensity for inflammation in STAT1 KO mice, but the phenotype is dependent on a trigger derived from a dysbiotic microbiome, suggesting a major role for tonic IFN in sculpting commensal diversity (*Pott and Stockinger, 2017*; *Kotredes et al., 2017*). How IFN can regulate the microbial population in the gut is still under investigation; however, we hypothesize that this could be achieved in a number of ways. (1) IFN could control gut microbiota through its multiple effects on the immune system, such as by modulating macrophage-mediated bacterial clearance (*Kumaran Satyanarayanan et al., 2019*). (2) TLR recognition of commensal derived products has been shown to be essential for intestinal epithelial homeostasis (*Rakoff-Nahoum et al., 2004*) since a number of TLR and TLR adapters are IFN inducible (*Siren et al., 2005*), it is likely that tonic IFN favors this process. (3) IFN promotes intestinal epithelial barrier function by enhancing epithelial cell differentiation (*Kotredes et al., 2017*), potentially protecting against bacterial invasion. (4) IFN has been suggested to have direct antimicrobial properties (*Kaplan et al., 2017*), although these effects may be only indirectly STAT1 dependent, for instance, through a positive feedback loop whereby STAT1-dependent signals amplify IFN expression (*Marie et al., 1998*). (5) IFN alters the expression of hundreds of target genes, possibly rewiring cell metabolism and altering metabolite output (*Fritsch and Weichhart, 2016*), thereby modifying the gut microbiota microenvironment. Interestingly, microbiota dysbiosis has also been associated with human diseases where the IFN system is perturbed, such as systemic lupus erythematosus (*Silverman et al., 2019*).

We found that the gut microbiota of STAT1-deficient mice is depleted of *Bacteroidales S24-7*, which has been described as a beneficial commensal in a mouse model of preclinical inflammatory arthritis (*Rogier et al., 2017*). This depletion was accompanied by a concomitant expansion of other families of bacteria, among them *Bacteroidales Prevotellaceae*, in particular species closely related to *P. heparinolytica*. These same changes in the composition of the microbiome were noted in GR STAT2 DKO mice, which phenocopied the inflammatory disease characteristics of STAT1 KO mice. However, these changes were not observed in mouse strains with only partial impairment of IFN signaling that did not phenocopy STAT1 KO. Of note, a number of *Prevotella* sp., including *P. copri, P. bivia, P. heparinolytica, P. nigrescens*, and *P. intermedia*, has been found to drive inflammation through induction of IL-17. *Prevotella* sp. have also been tied to a number of human chronic inflammatory diseases, such as new-onset rheumatoid arthritis (NORA), periodontitis, asthma, and bacterial vaginosis (*Scher et al., 2013*; *Larsen, 2017*; *Si et al., 2017*; *Lopes et al., 2020*). Nevertheless, we cannot rule out a possible contribution of additional bacterial species or of the commensal virome in the observed phenotype (*Neil and Cadwell, 2018*).

Our data raise the question of whether the altered microbiome of STAT1 KO mice is sufficient to trigger inflammatory disease. Since the microbiome of mice is readily transferred by cohabitation due to coprophagy (*Soave and Brand, 1991*; *Caruso et al., 2019*), we have cohoused STAT1 KO mice with WT mice but observed no alteration in the phenotypes of either strain. In addition, we maintained the STAT1 KO colony by interbreeding heterozygous mice, resulting in litters of mixed genotypes, an effective method of allowing redistribution of microbiome composition (*Robertson et al., 2019*). However, only homozygous KO offspring in these mixed cohorts developed inflammatory

disease. We speculate that competent IFN signaling in WT or heterozygous mice is sufficient to prevent dysbiosis.

The susceptibility of STAT1 KO mice to development of IBD is consistent with the observation that human patients carrying a hypomorphic allele of *STAT1* develop symptoms of colitis (*Thoeni et al., 2015*; *Sharfe et al., 2014*) and that many genes associated with IBD susceptibility are related to the IFN pathway (*Jostins et al., 2012*). IFN has been used as a therapy for IBD, but conflicting results have been reported concerning its efficacy in both ulcerative colitis and Crohn's disease, ranging from disease exacerbation to spectacular remissions (*Nikolaus et al., 2003*; *Ferre Aracil et al., 2016*). Further studies leading to a better understanding of the molecular mechanisms by which IFN controls gut microbiota to avert inflammatory pathologies will be invaluable for defining the set of patients and the timing where IFN therapy could be beneficial.

The role of IFNs in curbing overexuberant IL-17-driven inflammation by modulating the microbiota could be of clinical importance in a number of other diseases, including cancer (*Zhao et al., 2020*). For instance, STAT1 functions as a tumor suppressor through multiple mechanisms (*Meissl et al., 2017*) prevention of inflammation through modulation of the microbiota may provide another facet of its barrier to tumorigenesis. In sum, a delicate balance involving critical homeostatic IFN signaling aids in maintaining a healthy microbiome and taming systemic inflammation. Recent reports have emphasized the importance of gut microbiota on other organs, such as the central nervous system and the lungs (*Blacher et al., 2019*; *Budden et al., 2017*), suggesting that IFN may have a more far-reaching influence on overall human health and homeostasis through microbiome modulation than currently envisioned.

## Materials and methods

### Mice

The generation of *Stat1⁻/⁻*, *Ifnar⁻/⁻*, *Ifngr⁻/⁻*, *Stat2⁻/⁻*, and *Il17ra⁻/⁻* mice has been previously described (*Durbin et al., 1996*; *Müller et al., 1994*; *Huang et al., 1993*; *Park et al., 2000*; *Ye et al., 2001*). Mice were interbred to generate doubly or triply gene-deficient mice, as indicated, and genotypes were monitored by gene-specific PCR assays. All strains with the exception of *Stat1⁻/⁻* were maintained as homozygous mutant animals. The *Stat1* colony was maintained by harem interbreeding of *Stat1* heterozygous mice, and WT and *Stat1⁻/⁻* mice produced by this cross were compared. Except where indicated in specific experiments that comparisons involved littermates, WT and *Stat1⁻/⁻* offspring were pooled from several litters for experimental procedures. Mice were monitored for disease by inspection of coat condition, posture, enlarged peritoneal cavity indicative of splenomegaly, and peripheral blood counts using a Hemavet 950FS (Drew Scientific) on blood collected from the submandibular vein. All animals used in these experiments were maintained in a single dedicated room of a specific pathogen-free vivarium at NYU Grossman School of Medicine. All work with experimental animals was in accordance with protocols approved by the NYU Langone Health Institutional Animal Care and Use Committee, using animals 3–6 months of age. When indicated, animals were given an antibiotic cocktail made of 1 g/L each of ampicillin (Crystalgen), neomycin sulfate (Enzo), and metronidazole (Fisher) and 0.5 g/L of vancomycin hydrochloride (Fisher) in the drinking water, as previously described (*Rakoff-Nahoum et al., 2004*). Where noted, mice were injected weekly with 150 mg/kg of 5-fluorouracil (5-FU) for survival studies. Where indicated, mice were given 3 % dextran sulfate sodium (DSS) (M.W. 36.000–50.000, ICN) in drinking water for 7 days, changing the water every 2 days.

### Scoring of disseminated intestinal bacteria

After sacrifice, liver, mesenteric lymph node, and spleen lysates were plated on non-selective blood agar plates after serial dilution. Results are reported as cfu/organ.

### Histological scoring

Postmortem, the entire colon was removed, from the cecum to the anus, and the colon length was measured as a marker of inflammation. The entire colon was fixed in 10 % formalin, and paraffin sections were stained with hematoxylin and eosin (H&E). Histological scoring was performed in a blinded fashion by a pathologist, with individual scores for inflammatory cell infiltration severity (1: minimal, 2: mild, 3: moderate) and extent (1: expansion into mucosal tissue, 2: expansion to both

mucosal and submucosal tissues, 3: mucosal, submucosal and transmural invasion) as well as epithelial damage (1: minimal hyperplasia, 2–3: mild hyperplasia and 3–4: moderate hyperplasia extending to up to 50 % of epithelium) and goblet cell loss (scored 1–4 depending on severity) (*Erben et al., 2014*).

## Flow cytometry analysis

Single-cell suspensions were derived from mechanical disruption of mouse bone marrow and spleen in PBS supplemented with 2 % FCS (Sigma). For FACS analysis of spleen, bone marrow, and peripheral blood, red blood cells were lysed with Pharm Lyse Lysing buffer according to the manufacturer's instructions (BD Life Sciences). For progenitor staining, bone marrow and spleen cells were incubated with biotin-conjugated antibodies against committed lineage epitopes for 45 min on ice, washed once, and incubated with streptavidin and primary antibodies overnight on ice. For mature cell staining, cells were treated with Fc block for 5 min on ice, washed once, and incubated with primary antibodies for 45 min on ice. The antibodies used in this study are listed in *Supplementary file 1*. BD LSRII or BD FACSymphony (BD Life Sciences, San Jose, CA) were used for cell acquisition, and data were analyzed using FlowJo software (Treestar, Ashland, OR).

## Cytokine intracellular staining

Peripheral blood cells were plated after red blood cell lysis in DMEM 10 % serum containing 0.2% cell stimulation cocktail (eBiosciences) for 4 hr at 37 °C. Next, cells were collected, washed once, treated with Fc block for 5 min on ice, and stained with surface antibodies for 30 min on ice. Then, cells were washed, fixed in 2 % PFA for 15 min at room temperature, washed again, and permeabilized with 0.5 % saponin for 10 min at room temperature. Cells were then treated with Fc block diluted in 0.5 % saponin for 5 min and stained with intracellular antibodies diluted in 0.5 % saponin for 30 min at room temperature. Then cells were washed and incubated with 0.5 % saponin for an additional 15 min and analyzed by flow cytometry. The antibodies used in this study are listed in *Supplementary file 1*. BD LSRII (BD Life Sciences, San Jose, CA) was used for all analyses, and data were analyzed using FlowJo software (Treestar, Ashland, OR). Gating strategies are shown in *Figure 1—figure supplement 1*.

## Bone marrow and spleen transplant assays

$2 \times 10^6$ bone marrow or spleen cells harvested from donor mice of appropriate genotypes were intravenously injected in the retro-orbital sinus of near-lethally (~LD80) irradiated ($2 \times 5.7$ Gy) CD45.1$^+$ 8-week-old recipients. Donor spleens were depleted of T cells prior injection in recipient mice. Briefly, $1 \times 10^8$ cells were incubated with anti-CD8-Biotin and anti-CD4-Biotin for 15 min on ice. After one wash, cells were incubated with streptavidin-conjugated Magna beads (Biolegend) for 15 min on ice. The resulting T cell-depleted samples were injected in irradiated recipients. After 6 weeks, $2 \times 10^6$ whole bone marrow or spleen cells from primary recipient mice were transplanted into lethally irradiated CD45.1$^+$ secondary recipient mice. Irradiated mice were given water supplemented with antibiotics for 4 week. After 4 months, secondary recipients were analyzed for lineage distribution.

## Colony formation unit assay (methylcellulose)

Total bone marrow and spleen cells from WT and STAT1 KO mice (6–8 weeks) were plated in methylcellulose medium (Methocult M3434, Stem Cell Technologies). Cells were seeded at 20,000 total bone marrow cells and 100,000 total spleen cells per replicate. Colony forming units were enumerated using a Zeiss Axio Observer microscope and re-plated (2000 cells/replicate) every 7–10 days.

## Multiplex cytokine analysis

Cytokines and chemokines in serum were measured by the ProcartaPlex assay (mouse ProcartaPlex panel 1A, Invitrogen), performed on a Luminex 200 machine (Luminex). A panel of 36 cytokines and chemokines were analyzed: CXCL1 (C-X-C motif chemokine 1), CXCL2, CXCL5, CXCL10, CCL2 (C-C motif ligand 2), CCL3, CCL4, CCL5, CCL7, Eotaxin, IFN-α (interferon-α), IFN-γ, IL-1α (interlukin-1α), IL-1β, IL-2, IL-3, IL-4, IL-5, IL-6, IL-9, IL-10, IL-12p70, IL-13, IL-15/IL-15R, IL-17A, IL-18, IL-22, IL-23, IL-27, IL-28, IL-31, G-CSF (granulocyte colony-stimulating factor), GM–CSF (granulocyte-macrophage colony-stimulating factor), M-CSF (macrophage colony-stimulating factor), LIF (leukemia inhibitory factor), and TNF-α (tumor necrosis factor-α), according to the manufacturer's instructions. Cytokine concentrations were determined from the appropriate standard curves of known concentrations of

recombinant mouse cytokines and chemokines to convert fluorescence units to concentrations (pg/mL). Each sample was run in duplicate, and the mean of the duplicates was used to calculate the measured concentration.

## Gut microbiome analysis

### Gut microbiota sampling and DNA isolation

Mice fecal pellets were collected from individual mice in each group and stored immediately at −80 °C. All animals were aged-matched females, and each genotype was housed separately prior to fecal collection. DNA from fecal pellets was isolated using the Power Soil Kit (Qiagen), according to the manufacturer's instructions, and was stored at −80 °C prior to library preparation.

### 16S rRNA gene sequence analysis

The 16 S ribosomal RNA (16 S rRNA) V4-region was amplified by using the primers F515-R806, and the products for each fecal sample library were sequenced on a MiSeq instrument (Illumina), as previously described (*Azzouz et al., 2019*). DNA sequences were analyzed using QIIME two as previously described (*Caporaso et al., 2010*; *Bolyen et al., 2019*). Operational taxonomic units (OTUs) and amplicon sequence variants (ASVs) were obtained using the Dada2 plug-in and assigned using the GreenGenes database to obtain a taxonomy table.

For bacterial community visualization, R package Phyloseq was used to calculate the α-diversity index. Shannon index, Simpson index, and observed ASV abundance were used to estimate the community evenness and richness. Kruskal–Wallis test was used to obtain the overall p-value of the α-diversity index between groups. Phylogenetic analysis was performed using MegAlign Pro (DNA Star, Madison, WI).

### Murine T-cell isolation and culture

$CD4^+CD62L^+$ naïve T cells were isolated using CD4-negative enrichment kits (#19852, Stemcell Technologies, Vancouver, Canada) and CD62L microbeads (#130-049-701, Miltenyi Biotec, San Diego, CA) and confirmed to be >95% pure by flow cytometry. These cells were cultured on 96-well plates pre-coated with anti-CD3 and anti-CD28 (3 μg/ml each for $T_H17$ cultures, 1 μg/ml for $T_{reg}$ cultures). Cells were cultured in DMEM supplemented with neutralizing antibodies against IL-12, IFN-γ and IL-4 (clones C17.8, XMG1.2, and 11B11, 10 μg/ml each, BioXcell, West Lebanon, NH). $T_{reg}$ and $T_H17$ cultures were supplemented with TGF-β and IL-6 (Peprotech, Rocky Hill, NJ) as indicated. $T_{reg}$ cultures were fed with equal volume of IL-2-supplemented media (10 ng/ml, Peprotech) at day 2, split 1:2 into IL-2-supplemented media at day 3 and analyzed at day 4. $T_H17$ cultures were treated similarly except no IL-2 was added.

Five hours prior to analysis, $T_H17$ cultures were re-stimulated with PMA and ionomycin (50 and 500 ng/mL, respectively, Sigma-Aldrich, St. Louis, MO) in the presence of Golgistop (BD Life Sciences, San Jose, CA). Cells were typically stained with LIVE/DEAD (Thermo Fisher) and anti-CD4-FITC (RM4-5, Biolegend, San Diego, CA) before being fixed and permeabilized using Foxp3 fixation/permeabilization buffers (eBioscience, San Diego, CA). Intracellular staining with anti-IL-17 and anti-FOXP3 (clones eBio17B7 and FJK-16s, eBioscience) was performed per manufacturer's instructions. Acquisition was performed on a FacsVerse (BD Life Sciences, San Jose, CA) and analyzed using Flowjo software (Treestar, Ashland, OR).

## Statistical analysis

Differences between experimental groups were analyzed for statistical significance using unpaired two-tailed Student's t-test or Mann–Whitney U test, where indicated. Data shown are representative of at least three independent experiments, with specific details shown in each figure legend. A $p<0.05$ was considered to be statistically significant.

## Acknowledgements

The authors thank L Hu and E Maucotel for expert technical assistance and G David, F Boccalatte, and J A Hall for helpful discussions, gift of reagents and support. The authors gratefully acknowledge assistance from the NYU Histopathology, Flow Cytometry, and Microscopy core laboratories, especially C

Loomis, M Cammer, and M Gregory. The authors thank Amgen, Inc.Inc, for the kind gift of IL-17Ra KO mice. This work was supported in part by National Institutes of Health grants R01AI28900 to DEL, R01AI133822 to SSW, and P50AR070591 to GJS, Lupus Research Alliance grant 579817 to DEL, and by the Laura and Isaac Perlmutter Comprehensive Cancer Center support grant P30CA016087 from the National Cancer Institute.

## Additional information

### Competing interests

Gisele V Baracho: Gisele Baracho is affiliated with BD Life Sciences. The author has no financial interests to declare. The other authors declare that no competing interests exist.

### Funding

| Funder | Grant reference number | Author |
|---|---|---|
| National Institutes of Health | AI28900 | Isabelle J Marié<br>David E Levy |
| National Institutes of Health | AI133822 | Stephanie S Watowich |
| National Institutes of Health | AR070591 | Gregg Silverman |
| National Institutes of Health | CA016087 | Pratip Chattopadhyay |
| Lupus Research Alliance | 579817 | David E Levy<br>Isabelle J Marié |

The funders had no role in study design, data collection and interpretation, or the decision to submit the work for publication.

### Author contributions

Isabelle J Marié, Conceptualization, Data curation, Formal analysis, Funding acquisition, Investigation, Methodology, Project administration, Supervision, Validation, Visualization, Writing – original draft, Writing – review and editing; Lara Brambilla, Investigation, Methodology, Supervision, Validation, Visualization, Writing – original draft, Writing – review and editing; Doua Azzouz, Formal analysis, Investigation, Methodology; Ze Chen, Conceptualization, Data curation, Investigation, Methodology, Supervision, Visualization; Gisele V Baracho, Conceptualization, Data curation, Formal analysis, Investigation, Methodology, Visualization, Writing – original draft, Writing – review and editing; Azlann Arnett, Investigation, Methodology, Visualization, Writing – review and editing; Haiyan S Li, Data curation, Formal analysis, Investigation; Weiguo Liu, Investigation, Methodology; Luisa Cimmino, Conceptualization, Data curation, Methodology, Visualization, Writing – review and editing; Pratip Chattopadhyay, Investigation, Methodology, Supervision; Gregg Silverman, Conceptualization, Data curation, Formal analysis, Funding acquisition, Investigation, Supervision, Writing – review and editing; Stephanie S Watowich, Conceptualization, Investigation, Methodology, Supervision, Writing – review and editing; Bernard Khor, Data curation, Formal analysis, Investigation, Methodology, Supervision, Visualization, Writing – review and editing; David E Levy, Conceptualization, Data curation, Funding acquisition, Investigation, Methodology, Project administration, Supervision, Writing – original draft, Writing – review and editing

### Author ORCIDs

Isabelle J Marié (ORCID) http://orcid.org/0000-0002-3091-7624
Stephanie S Watowich (ORCID) http://orcid.org/0000-0003-1969-659X
Bernard Khor (ORCID) http://orcid.org/0000-0003-4689-5092
David E Levy (ORCID) http://orcid.org/0000-0002-7320-7788

### Ethics

All animals used in these experiments were maintained in a single dedicated room of a specific pathogen-free vivarium at NYU Grossman School of Medicine. All work with experimental animals was in accordance with protocols approved by the NYU Langone Health Institutional Animal Care and Use Committee (IACUC ID: IA16-01579).

### Decision letter and Author response

Decision letter https://doi.org/10.7554/eLife.68371.sa1
Author response https://doi.org/10.7554/eLife.68371.sa2

## Additional files

### Supplementary files

• Supplementary file 1. List of antibodies used for flow cytometry.
• Supplementary file 2. Summary of mutant phenotypes.
• Transparent reporting form

### Data availability

Sequencing data related to microbiome analysis have been deposited in Dryad at http://dx.doi.org/10.5061/dryad.b5mkkwhcv. Source data files for other figures are provided with the manuscript.

The following dataset was generated:

| Author(s) | Year | Dataset title | Dataset URL | Database and Identifier |
|---|---|---|---|---|
| Brambilla Lara, Azzouz Doua, Chen Ze, Marie IJ | 2021 | Tonic interferon restricts pathogenic IL-17-driven inflammatory disease via balancing the microbiome, Dryad, Dataset | http://dx.doi.org/10.5061/dryad.b5mkkwhcv | Dryad Digital Repository, 10.5061/dryad.b5mkkwhcv |

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
