## [Decision Letter]

**Acceptance summary:**

This manuscript identifies an essential role for tonic interferon signaling in preventing auto-inflammatory bowel disease in mice. The authors show that the absence of interferon signaling leads to alterations in the normal composition of the gut microbiota and the inappropriate expansion of a class of inflammatory cells producing interleukin-17. Genetic deletion of the interleukin-17 receptor or antibiotic treatment ameliorated the inflammatory disease, suggesting a causative role for these factors in the disease.

**Decision letter after peer review:**

Thank you for submitting your article "Tonic interferon restricts pathogenic IL-17-driven inflammatory disease via balancing the microbiome" for consideration by *eLife*. Your article has been reviewed by 3 peer reviewers, one of whom is a member of our Board of Reviewing Editors, and the evaluation has been overseen by Tadatsugu Taniguchi as the Senior Editor. The reviewers have opted to remain anonymous.

The reviewers have discussed their reviews with one another, and the Reviewing Editor has drafted a consolidated summary (below). Overall, all three reviewers were interested in the findings and can appreciate their importance and impact for the field. We thus encourage submission of a revised manuscript, assuming that the essential revisions can be satisfactorily addressed.

Essential revisions:

1. The relationship between the various observed phenotypes (dysbiosis, splenomegaly, IL17 production, etc) in the STAT1-deficient mice is not clear. If loss of IFN signaling is sufficient to produce gut dysbiosis, then this dysbiosis should be seen in the STAT1/IL-17 double-deficient mice (already generated and analyzed by the authors). This analysis, as well as a characterization of the inflammatory state of these mice (i.e., cytokines), should be conducted to distinguish whether dysbiosis/inflammation is upstream or downstream of IL-17.

2. In order to establish that indeed the pheontype of the STAT2/IFNGR mice phenocopies the STAT1 deficient mice, the authors should show that antibiotic treatment also rescues the STAT2/IFNGR deficient mice.

3. Figure 3A. It appears that the induction of IL17A in STAT1 KO mice is not statistically significant. Presumably this is due to a low number of analyzed mice? Since the spontaneous induction of IL17 in STAT1 KO is a major claim of the manuscript, this result needs to be statistically significant, so further repeats should be done. Moreover, in this figure and elsewhere when results from several mice are shown, the individual mice should be shown as independent data points (instead of just showing mean and SD) so that the spread and distribution of the data can be assessed. Is the data normally distributed? (if not, the student t-test may not be appropriate).

4. p10, the authors conclude "disease observed in STAT1 KO mice is not fully cell intrinsic". It is not clear how this conclusion is reached from the experiment performed (generation of a STAT1KO>WT chimera). Perhaps the authors mean that STAT1 does not function solely in hematopoietic cells? As a molecule required for receptor signal transduction, it would surprising if STAT1 did not act cell-intrinsically, but this would have to be shown in a mixed BM chimera (e.g., STAT1KO+WT>STAT1KO). Ideally the authors would report the full set of chimera experiments (i.e., including WT>STAT1 and STAT1>STAT1) as this would help address in what cell types STAT1 functions to suppress inflammatory disease. In particular, given the known ability of type III IFNs to act on intestinal epithelial cells, some discussion of the cell types that respond to IFN to regulate inflammation seems important.

5. p.20 It is claimed "Bacteria were observed mainly in the liver in STAT1 KO mice and to a lesser extent in the mesenteric lymph nodes and very rarely in the spleen (Figure 5D)". However, the data in the figure just show a combined "score". Why aren't CFU/organ reported? This would be preferable, but at a minimum scores for each separate organ should be reported to support the claim made in the text.

6. The legend for figure 5 does not correspond to the figure panels.

7. p.23 please use "sex-matched" instead of "gender-matched" as gender is a socially constructed identity (not a biological trait) and is not usually ascribed to mice.

8. A discussion of the previously reported microbiota/IFN relationship is missing (e.g. PMID: 32380006, PMID: 22705104,PMID: 22749822, PMID: 31269444).

9. The authors should reconsider the use of term "resolution of dysbiosis by antibiotic treatment". I recommend stating what the antibiotics actually did, i.e., reduce the microbial load.

10. The conclusion that STAT1 KO blood showed a compensatory loss of B lymphocytes is not supported by the data since It is impossible to draw such a conclusion based on a graph displaying relative frequencies of cells in a given tissue. This claim must be based on cell cell numbers (which are not shown). In fact, in Figures 1E and F the increased neutrophilia in STAT1 KO spleens caused relative changes in the % of splenocytes that B cells represent, but the actual number of B cells was actually increased, meaning there was no "compensatory LOSS of B cells". The same principle applies to the discussion of the data presented in 1G. The authors either need to show cell numbers for all these populations or delete statements claiming reductions based on relative abundance measurements.

11. It is important that the authors state the age of ALL mice used throughout the paper. The statement that "animals developed splenomegaly irrespective of age" is difficult to interpret. If the disease develops spontaneously, it seems that there would be a defined age/period of onset. At what age are the splenomegaly and the spontaneous colitis evident? Depending on the age of colitis onset, this likely impacts the microbiota differences detected in Figure 6. And if the inflammation occurs early, this would obviously impact the DSS results, and perhaps render the DSS results unnecessary.

12. If the WT splenocytes did not have enough HSCs to survive in/reconstitute the recipients (Figure 2), how were the recipients able to survive until the end of the experiment?

13. Please clearly state and address the similarities and differences with the previous Porpaczy et al. paper. Did they also observe a lack of disease in WT mice transplanted with STAT1 KO bone marrow? (if not, why not?) The Porpaczy et al. paper also showed that STAT1-deficient mice harbored malignant B cells which should have transferred with the bulk populations utilized in this study. Was this evident in the present study as well?

---

## [Author Response]

Essential revisions:1. The relationship between the various observed phenotypes (dysbiosis, splenomegaly, IL17 production, etc) in the STAT1-deficient mice is not clear. If loss of IFN signaling is sufficient to produce gut dysbiosis, then this dysbiosis should be seen in the STAT1/IL-17 double-deficient mice (already generated and analyzed by the authors). This analysis, as well as a characterization of the inflammatory state of these mice (i.e., cytokines), should be conducted to distinguish whether dysbiosis/inflammation is upstream or downstream of IL-17.

We have conducted additional microbiome sequencing and cytokine production analysis of STAT1-IL17Ra DKO mice (new Figure S6 and S7). These new data show that loss of STAT1 is the driver of the inflammatory phenotype, with IL17 acting as an effector ‘downstream’ of STAT1 loss.

2. In order to establish that indeed the pheontype of the STAT2/IFNGR mice phenocopies the STAT1 deficient mice, the authors should show that antibiotic treatment also rescues the STAT2/IFNGR deficient mice.

We provide new data showing that antibiotic treatment of STAT2-GR-DKO mice averts the inflammatory disease phenotype, similar to the results with STAT1-KO (new Figure S4).

3. Figure 3A. It appears that the induction of IL17A in STAT1 KO mice is not statistically significant. Presumably this is due to a low number of analyzed mice? Since the spontaneous induction of IL17 in STAT1 KO is a major claim of the manuscript, this result needs to be statistically significant, so further repeats should be done. Moreover, in this figure and elsewhere when results from several mice are shown, the individual mice should be shown as independent data points (instead of just showing mean and SD) so that the spread and distribution of the data can be assessed. Is the data normally distributed? (if not, the student t-test may not be appropriate).

At the reviewers’ suggestion, we reanalyzed cytokine production data using non-parametric statistics (Mann-Whitney) due to the data being not normally distributed. This analysis showed that IL17A production was statistically different between WT and STAT1 KO animals. In addition, we have analyzed cytokines in a new cohort of STAT1 KO mice along with IL17 KO and IL17 STAT1 DKO mice (new Figure S6), which again shows that the increase in IL17A abundance is statistically significant. As recommended by the reviewers, we have modified the figures where appropriate to indicate the values for each individual mouse, except for very complex figures, where doing so could lead to confusion. In all cases, however, raw data were uploaded as source data. Data sets that were not normally distributed have been analyzed by nonparametric statistics, as indicated.

4. p10, the authors conclude "disease observed in STAT1 KO mice is not fully cell intrinsic". It is not clear how this conclusion is reached from the experiment performed (generation of a STAT1KO>WT chimera). Perhaps the authors mean that STAT1 does not function solely in hematopoietic cells? As a molecule required for receptor signal transduction, it would surprising if STAT1 did not act cell-intrinsically, but this would have to be shown in a mixed BM chimera (e.g., STAT1KO+WT>STAT1KO). Ideally the authors would report the full set of chimera experiments (i.e., including WT>STAT1 and STAT1>STAT1) as this would help address in what cell types STAT1 functions to suppress inflammatory disease. In particular, given the known ability of type III IFNs to act on intestinal epithelial cells, some discussion of the cell types that respond to IFN to regulate inflammation seems important.

We apologize for our lack of clarity on the interpretation of the transplant data. The reviewers are absolutely correct that STAT1 likely acts in a cell intrinsic manner, and we have now clarified that the phenotype is not due solely to STAT1 acting in the hematopoietic compartment. As now better described in the text, transplantation of STAT1KO hematopoietic cells into WT recipients is not sufficient to trigger inflammation, presumably because IFNs are also acting in non-hematopoietic tissues. As suggested by the reviewers, we also now more fully discuss the known ability of type III IFNs to act on intestinal epithelial cells. To more fully evaluate the cell type-specific role of STAT1 will require analysis of conditional STAT1 knockout mice crossed with appropriate tissue-specific Cre mice, but such experiments are beyond the scope of this report.

5. p.20 It is claimed "Bacteria were observed mainly in the liver in STAT1 KO mice and to a lesser extent in the mesenteric lymph nodes and very rarely in the spleen (Figure 5D)". However, the data in the figure just show a combined "score". Why aren't CFU/organ reported? This would be preferable, but at a minimum scores for each separate organ should be reported to support the claim made in the text.

* *The figure has been modified to present the data as suggested by the reviewers.

6. The legend for figure 5 does not correspond to the figure panels.

The legend to Figure 5 has been modified to correctly describe the data in the figure.

7. p.23 please use "sex-matched" instead of "gender-matched" as gender is a socially constructed identity (not a biological trait) and is not usually ascribed to mice.

* *Modified as suggested.

8. A discussion of the previously reported microbiota/IFN relationship is missing (e.g. PMID: 32380006, PMID: 22705104,PMID: 22749822, PMID: 31269444).

We have expanded the discussion to more fully place the data in the context of the literature already published on microbiota/IFN relationships.* *

9. The authors should reconsider the use of term "resolution of dysbiosis by antibiotic treatment". I recommend stating what the antibiotics actually did, i.e., reduce the microbial load.

We thank the reviewer for this suggestion and now describe the action of antibiotics as requested, that is, reduction of the microbial load and averting the development of inflammation.* *

10. The conclusion that STAT1 KO blood showed a compensatory loss of B lymphocytes is not supported by the data since It is impossible to draw such a conclusion based on a graph displaying relative frequencies of cells in a given tissue. This claim must be based on cell cell numbers (which are not shown). In fact, in Figures 1E and F the increased neutrophilia in STAT1 KO spleens caused relative changes in the % of splenocytes that B cells represent, but the actual number of B cells was actually increased, meaning there was no "compensatory LOSS of B cells". The same principle applies to the discussion of the data presented in 1G. The authors either need to show cell numbers for all these populations or delete statements claiming reductions based on relative abundance measurements.

Again we thank the reviewers for pointing out our lack of clarity. We now describe all the changes observed in leukocytes correctly, either as a change in percentage or number, as appropriate, and we avoid making incorrect assumptions about compensation.

11. It is important that the authors state the age of ALL mice used throughout the paper. The statement that "animals developed splenomegaly irrespective of age" is difficult to interpret. If the disease develops spontaneously, it seems that there would be a defined age/period of onset. At what age are the splenomegaly and the spontaneous colitis evident? Depending on the age of colitis onset, this likely impacts the microbiota differences detected in Figure 6. And if the inflammation occurs early, this would obviously impact the DSS results, and perhaps render the DSS results unnecessary.

We have clarified in the text that the experimental animals ranged in age from 3-6 mo, with comparisons being made between animals of approximately the same age. It would be quite laborious, although of course possible, to identify the age of each mouse used for every data point. Moreover, we did not observe a temporal progression of disease during this time frame. As shown in the data from the DSS experiment, both WT and STAT1 KO mice showed more severe colitis after DSS treatment, with heightened inflammation on the mutant background.

12. If the WT splenocytes did not have enough HSCs to survive in/reconstitute the recipients (Figure 2), how were the recipients able to survive until the end of the experiment?

We thank the reviewers for requesting clarification on this point. As now more fully discussed, recipient mice were irradiated to ~LD80, such that the majority did not survive without transplantation. The few that survived displayed after transplantation with WT splenocytes displayed only recipient-derived leukocytes, which presumably was sufficient for survival. The goal of irradiation in this context is to create an available niche for engraftment of donor cells, not necessarily to eliminate all endogenous recipient cells, since the transplantations were syngeneic.

13. Please clearly state and address the similarities and differences with the previous Porpaczy et al. paper. Did they also observe a lack of disease in WT mice transplanted with STAT1 KO bone marrow? (if not, why not?) The Porpaczy et al. paper also showed that STAT1-deficient mice harbored malignant B cells which should have transferred with the bulk populations utilized in this study. Was this evident in the present study as well?

We have now more fully discussed a comparison of our results with those reported by Porpaczy et al. Both studies saw myeloid expansion in STAT1 KO mice that did not recur following transplantation of STAT1 KO bone marrow into WT recipients. A difference between the two studies is that Porpaczy et al. observed the emergence of malignant B cell clones, either following suppression of myelopoiesis or at extended times after transplantation. We have not reproduced those exact conditions. However, we also suspect that the B malignancy may relate to strain background differences (C57BL6 vs. Balb/c) and to the known susceptibility of Balb/c mice to B malignancies due to mutation of the p16INK4a tumor suppressor.